**communications** engineering

# Towards ubiquitous radio access using nanodiamond based quantum receivers
Qunsong Zeng [1,3], Jiahua Zhang[1,3], Madhav Gupta[1], Zhiqin Chu [1,2] ✉ & Kaibin Huang [1] ✉

The development of sixth-generation wireless communication systems demands innovative solutions to address challenges in the deployment of a large number of base stations and the detection of multi-band signals. Quantum technology, specifically nitrogen-vacancy centers in diamonds, offers promising potential for the development of compact, robust receivers capable of supporting multiple users. Here we propose a multiple access scheme using fluorescent nanodiamonds containing nitrogen-vacancy centers as nano-antennas. The unique response of each nanodiamond to applied microwaves allows for distinguishable patterns of fluorescence intensities, enabling multi-user signal demodulation. We demonstrate the effectiveness of our nanodiamonds-implemented receiver by simultaneously transmitting two uncoded digitally modulated information bit streams from two separate transmitters, achieving a low bit error ratio. Moreover, our design supports tunable frequency band communication and reference-free signal decoupling, reducing communication overhead. Furthermore, we implement a miniaturized device comprising all essential components, highlighting its practicality as a receiver serving multiple users simultaneously. This approach enables the integration of quantum sensing technologies into future wireless communication networks.

While the fifth-generation (5G) mobile technologies are being widely deployed to transform people's lifestyles, researchers have embarked on the development of the sixth-generation (6G) and the exploration of its killer applications[1]. In the 6G era, the requirement for ubiquitous radio access poses significant challenges for the implementation and management of wireless communication systems[2]. Among the most pressing issues is the deployment of a large number of base stations[3] due to their typically large sizes[4]. Traditionally, the number of antenna elements that can be integrated into a practically feasible antenna size is limited due to the required half-wavelength spacing between elements[4]. On the other hand, due to antennas' finite bandwidths, the detection of multi-band signals requires an array of costly radio frequency (RF) receivers, each tailored for a specific band[5]. To fully realize the potential of 6G networks by ensuring seamless connectivity across a wide range of applications, it is crucial to address the above issues, particularly by developing small, robust receivers capable of supporting access by multiple users.

Lately, quantum technology empowered signal receivers[6–9] have garnered strong interests due to their ability to replace or even surpass conventional "antennas" in the fields of communication[10,11], medicine[12,13], and navigation[14,15]. Compared to conventional metal antennas and newly developed electrical-optical systems[16–18], quantum receivers offer several advantages, including low-loss transmission, reduced power requirements, and high sensitivity for weak signal recovery. Among these, nitrogen vacancy (NV) centers in diamonds are notable for their exceptional attributes, including high sensitivity and accuracy[19–23], tunable working bandwidth[24–26], scalability[27], and adaptability to extreme conditions[28]. In contrast with traditional receivers that consists of a series of components such as RF antennas, analog filters, and mixers, NV center-based receivers mainly utilize optical components for detection and down-conversion, making them inherently resistant to electrical noise[29]. Despite the rapid advancement of quantum receivers employing diamonds with NV centers, their deployment remains restricted to rudimentary demonstrations, such as point-to-point (single-input-single-output) systems[19–28]. To tackle 6G challenges, it is now crucial to develop concurrently multi-accessible quantum receivers to support multiple users, thereby unleashing their full potential in the wireless communication domain.

Here we propose a feasible multiple access scheme using robust fluorescent nanodiamonds (FNDs) containing NV centers as nano-receivers (Fig. 1). The key concept revolves around the utilization of distinct features of each individual FND, in terms of the number, location, and orientation of spin defects. By exploiting the unique response of each FND to applied microwaves, we can achieve distinguishable patterns of multiple FNDs' fluorescence intensities. This approach, integrated with modulation techniques in communication, facilitates multi-user signal demodulation

[1]Department of Electrical and Electronic Engineering, The University of Hong Kong, Hong Kong, China. [2]School of Biomedical Sciences, The University of Hong Kong, Hong Kong, China. [3]These authors contributed equally: Qunsong Zeng, Jiahua Zhang. ✉e-mail: zqchu@eee.hku.hk; huangkb@eee.hku.hk

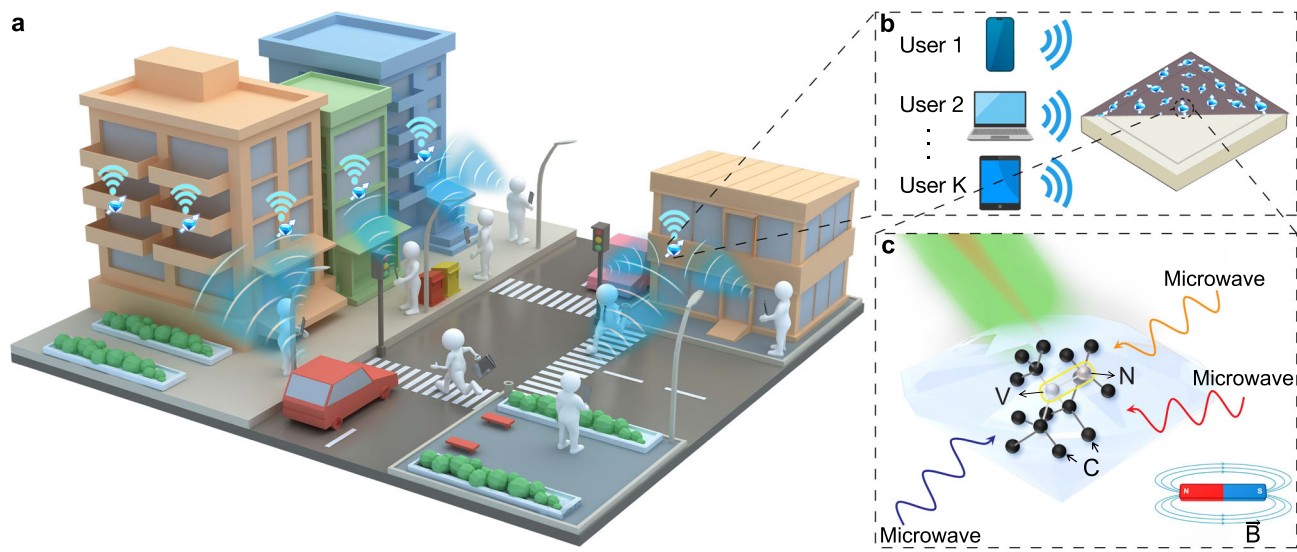

**Fig. 1 | The concept of nanodiamond based receivers for ubiquitous radio access.** **a** The 6G ubiquitous radio access scenario and the proposed deployment of quantum receivers. The approach leverages the unique properties of quantum systems, specially nitrogen-vacancy (NV) centers in fluorescent nanodiamonds (FNDs), to create highly sensitive and compact receivers. **b** These FND-receivers enable efficient and simultaneous detection of multiple radio signals from different users across a broad spectrum in the wireless communication system. **c** The NV center point defect in a diamond lattice is composed of a substitutional nitrogen atom (N) and an adjacent crystallographic vacancy (V). Black spheres represent carbon atoms (C). Upon optical excitation by a green laser, the population of the ground state can be read out by monitoring the intensity of the emitted fluorescence light. The working frequency band can be altered by applying an external magnetic field ($\vec{B}$).

based on varying fluorescence intensity. We demonstrated two transmitters simultaneously emitting microwaves containing uncoded digitally modulated information bits from different grayscale images, with each transmitter transmitting a total of 14,464 bits, to evaluate the performance of our FNDs-implemented receiver. The experiments measured a bit error ratio (BER) of 0.146% for two $28 \times 28$ uint-8 handwritten digit images and 0.439% for two $32 \times 32$ uint-8 face images using frequency modulation, while amplitude modulation resulted in a BER of zero. The corresponding bit error probability (BEP) for frequency modulation was calculated as $(0.0031 \pm 0.0009)$, while for amplitude modulation, it was less than 0.000069. Moreover, by detuning the spin resonance frequencies of NV centers with applied external static magnetic fields, our approach can also support tunable frequency band for communication. Exposed to such magnetic field, the heterogeneity of NV centers in FNDs allows for a reference-free design, decoupling the multiple users' signals by observing the response of different FNDs to different frequencies, thereby reducing communication overhead. Based on conservative estimates, our system can support up to five users with a field of view of $75\,\mu m \times 75\,\mu m$ and magnetic field gradients of approximately $0.023\,G/\mu m$, resulting in a utilization ratio of 11.1% (5 out of 45) for FNDs. As a proof-of-concept demonstration, we show the transmission capability using 2 out of 5 users, achieving a BER of 0.0657%. Furthermore, we have constructed a miniaturized FND-receiver device comprising all essential components to show its immediate potential as a practical receiver supporting multiple users simultaneously.

## Results

### Working principles of a diamond receiver

The principle underlying the diamond receiver, which utilizes the diamond with NV centers as an antenna, relies on the optically detected magnetic resonance (ODMR). When the microwave frequency is in resonance with the ground state energy gap between the $|0\rangle$ spin state and the two degenerate $|\pm 1\rangle$ spin states (Supplementary Fig. 1a), NV centers experience a drop in fluorescence intensity. This drop is characterized by a Lorentzian peak at approximately 2.87 GHz, as shown in Supplementary Fig. 1b (yellow curve). The region around this peak enables the conversion of RF signals into fluorescence signals (Fig. 2a). While a large ensemble of NV centers with random orientations shows a contrast of only a few percent, the FNDs used

in our study generally exhibit a contrast of around 10%. Based on this contrast, the sensitivity of our FNDs is $0.735\,\mu T \cdot Hz^{-1/2}$ when ODMR is applied for detecting the magnetic component of the incident electromagnetic field. A comprehensive discussion on the sensitivity and contrast can be found in Supplementary Note 4. By exploiting the distinct properties of ODMR, it becomes feasible to detect signals, especially those employing prevalent amplitude and frequency modulations (Fig. 2b–d). The contrast of ODMR spectrum exhibits an inverse linear correlation with the power of the incident microwave signal (Fig. 2e), allowing for the detection of the amplitude modulated signals. Moreover, the fluorescence intensity to the right of the Lorentzian peak interval is roughly proportional to the frequency of the incoming microwave signal (Fig. 2f), enabling the effective use of frequency modulation within this specific range. We tested one dimensional audio signals and observed that residual levels in our system are below 0.4% for both amplitude and frequency modulations (Fig. 2h, i). This indicates that the diamond receiver based on NV centers can effectively detect and demodulate amplitude and frequency modulated signals with high fidelity. Furthermore, we examined the diamond receiver's capability in terms of joint amplitude and frequency modulation in a digital communication system. The fluorescence intensity is influenced by both the power and frequency of the incident microwave signal, as evidenced by the observation of four distinguishable combinations (Fig. 2g). As shown in Fig. 2j, our analysis of the audio signal showed an insignificant residual, with a maximum absolute value of 0.5%, which indicates that the diamond receiver can effectively handle complex modulation schemes with minimal error. Through these experimental results, the diamond receiver has exhibited exceptional performance in detecting both amplitude and frequency-modulated microwave signals.

### NV centers based multiple access system

The principle of the diamond receiver described above is primarily designed for the point-to-point communication, i.e., a single user scenario, but is not sufficient for serving multiple users simultaneously. In most communication networks, an access point is designed to accommodate multiple users. Hence, for practical implementation in such contexts, a multiple access system needs to be developed to cater to multiple users concurrently. Beyond the use of bulk diamond, an FND containing NV centers can serve

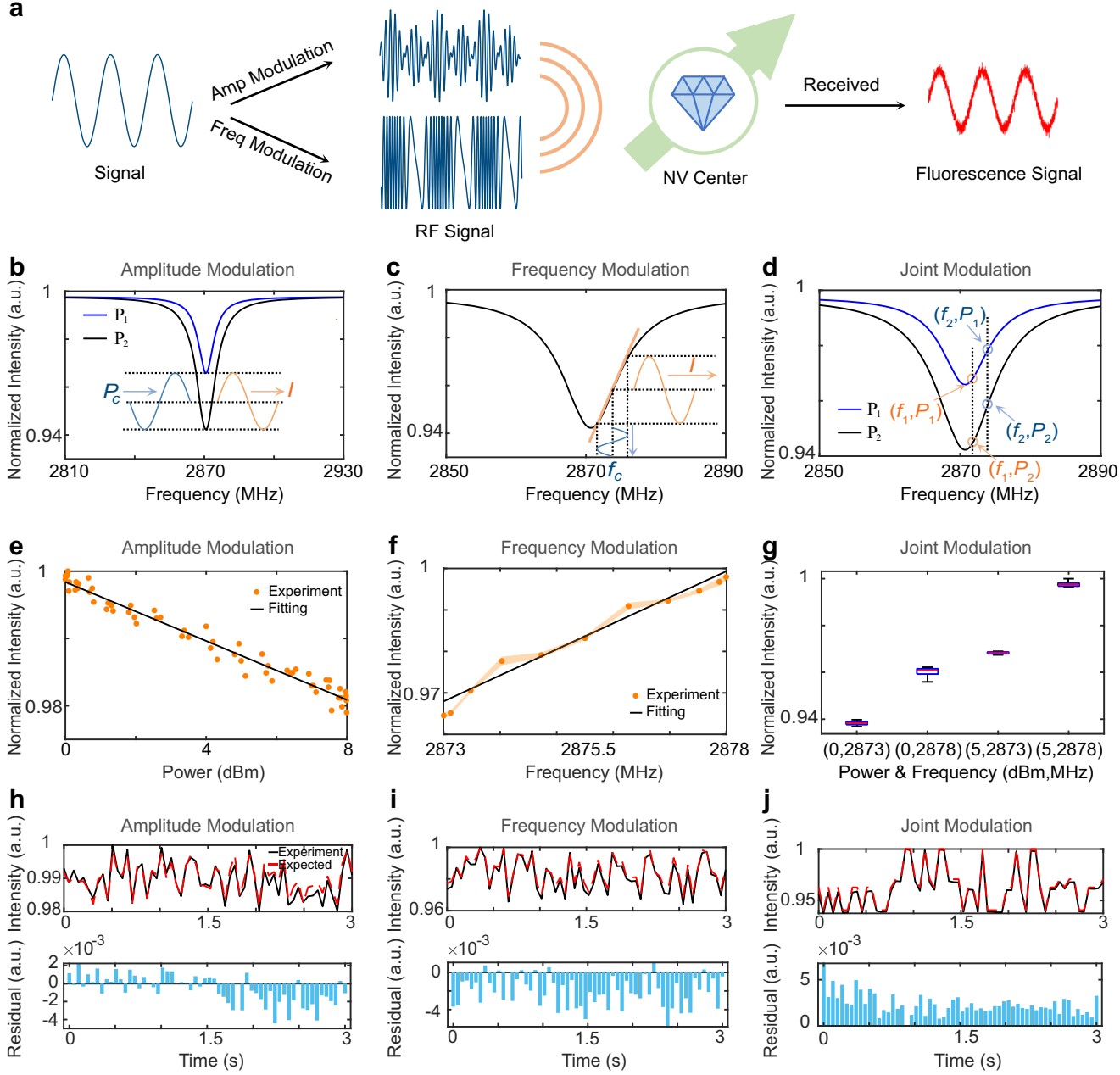

**Fig. 2 | Working principles of diamond NV center-based receiver. a** NV center-based RF signal detection: Converting amplitude or frequency modulated microwaves into fluorescence intensities. **b** Amplitude demodulation illustrates optically detected magnetic resonance (ODMR) near 2.87 GHz with transmission powers $P_1 < P_2$. **c** Frequency demodulation places the carrier frequency on the slope, mapping frequency changes onto fluorescence intensities. **d** Joint demodulation shows four unique combinations of frequencies and power levels. **e** Amplitude modulation reveals an inverse relationship between fluorescence intensity and microwave power. **f** Frequency modulation demonstrates a direct correlation between fluorescence intensity and microwave frequency. **g** Results of 30 experimental repeats. Joint amplitude-and-frequency modulation presents four distinct fluorescence intensity states. **h–j** Comparison of demodulated fluorescence intensities with expected audio signals, accompanied by respective residuals in histograms, for (**h**) amplitude, (**i**) frequency, and (**j**) joint modulations.

as the antenna in the aforementioned receiver, albeit with a slight compromise in signal-to-noise ratio (SNR). When numerous FNDs are randomly distributed on a cover glass, they serve as multiple anisotropic antennas, enabling them to function as a multi-antenna receiver in the multiple access system. This setup facilitates the demultiplexing of signals from different users. This capability stems from three key facts: microwaves from different users encounter distinct wireless channels; spins of FNDs possess varying orientations; and the number of NV centers in FNDs differs. As a result, FNDs exhibit diverse responses to different combinations of incident microwaves, enabling the system to distinguish and process multiple user signals simultaneously.

The operation of a multiple access system with an FND-receiver is outlined as follows. The bit stream of each user is composed of reference bits and data bits, as depicted in Fig. 3a. The reference bits are devised to realize all possible combinations. To enable simultaneous detection of signals from multiple users, the transmission of reference bits takes precedence. These bits generate reference fluorescence images at the FND-receiver's end. Subsequently, the received fluorescence images from data bits are compared to these reference images for signal demultiplexing. For proof-of-concept, we consider a two-user scenario in our experiments. Each user transmits an image message, with each pixel represented using 8 bits in uint8 format. For digital frequency modulation, we assign bits 0 and 1 to frequencies

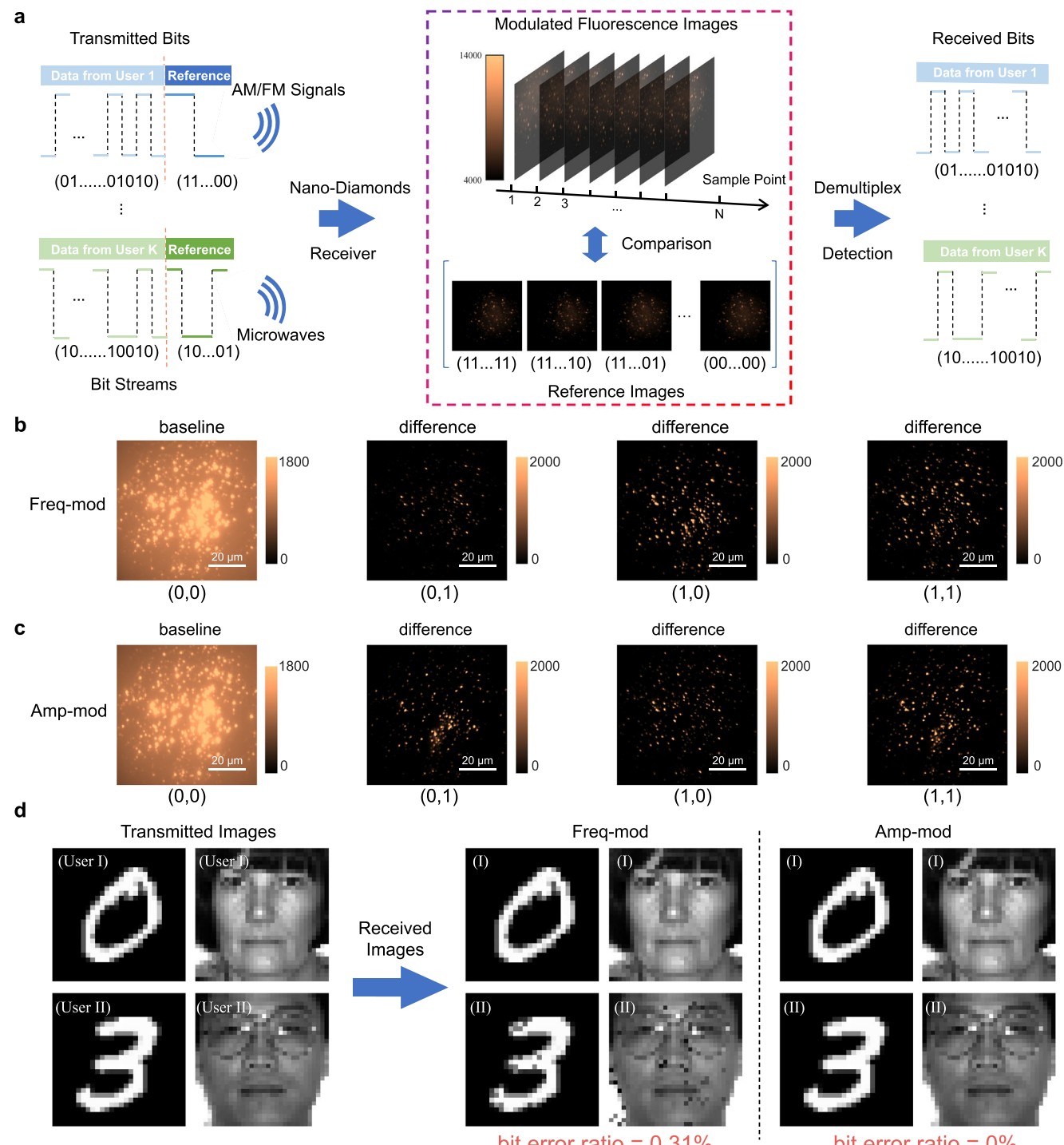

**Fig. 3 | Experimental demonstration of a multiple access system. a** Schematic illustration of the fluorescent nanodiamond (FND)-receiver for supporting a multiple access system. Users transmit reference bits followed by data transmission, resulting in reference images, i.e., fluorescence intensity matrix, composed of fluorescence intensity measurements of a region of the plate containing randomly placed FNDs. During data bits transmission, the fluorescence images are compared to the reference images to be demultiplexed using the minimum mean square error (MSE) criterion. **b** Reference images obtained in the frequency modulation experiment. The baseline corresponds to the fluorescence intensity matrix when both users simultaneously transmit bit "0". The other three images represent the remaining three-bit combinations, with their differences from the baseline. **c** Reference images acquired in the amplitude modulation experiment, with the same interpretation as described. **d** Experiment involving the transmission of (Left) two digit images from MNIST Database[30] and (Right) two subsampled face images from Yale Face Database[31]: Image messages transmitted by two users, image messages received during the frequency modulation experiment, and image messages received during the amplitude modulation experiment.

2870 MHz and 2900 MHz, respectively. The resultant fluorescence images, induced by the received pairs of reference bits, serve as the basis for bit-pair detection. Due to the diversity of FNDs' responses, different bit-pairs generate varied reference fluorescence images. This concept is illustrated by the differences between the reference fluorescence images displayed in Fig. 3b.

Following the transmission of reference bits, data bits – the bits of the transmitted messages – are sent. To identify the data bit-pair at each sampling point, we compare the received fluorescence images with the reference images, pinpointing the bit-pair that provides the minimum mean squared error (MSE) at each sampling point. This detection process can be

mathematically expressed as $(x, y)_t^\star = \text{argmin}_{(x,y)}||\mathbf{I}_t - \mathbf{R}(x,y)||_F^2$, where $(x, y)_t^\star$ represents the detected optimal bit-pair at the $t$-th sampling point during data transmission, $|| \cdot ||_F$ refers to the Frobenius norm, $\mathbf{I}_t$ denotes the received fluorescence image at the $t$-th sampling point, and $\mathbf{R}(x,y)$ is the fluorescence image of the corresponding reference bit-pair $(x, y)$. This detection process is conceptually similar to a lookup table, where matching operations are performed to identify the correct bit-pair. By detecting the bit-pair at each sampling point, we can recover the bit for each image message, ultimately reconstructing the two transmitted image messages at the FND-receiver's end. We tested the transmission of two types of image message: handwritten digit images[30] and face images[31], with the latter requiring a higher communication overhead. Figure 3d illustrates the transmission results for both of them. For frequency-modulated transmission, the BER for handwritten digit images is 0.146%, while for face images, the BER is slightly higher at 0.439%. The associated BEP, taking into account its uncertainty, is estimated at (0.0031 ± 0.0009) at a 95% confidence level. These results highlight the effectiveness and accuracy of the proposed FND-receiver in distinguishing and demodulating the transmitted signals from multiple users in a multiple access system. Moreover, for amplitude-modulated transmission, the system operates at a single frequency of 2870 MHz. For one user, bits 0 and 1 are modulated to power levels of -15 dBm and -7 dBm, while for the other user, they are modulated to power levels of -15 dBm and 0 dBm, respectively. The remaining procedures mirror those of the frequency modulation case. The resultant reference images are depicted in Fig. 3c, and the final recovered images are shown in Fig. 3d, with the BERs being zero for both sets of image message. This indicates that the BEP is less than 0.000069. Furthermore, the digital transmission with joint amplitude and frequency modulation extends the principles and designs for frequency modulation and amplitude modulation discussed above. The results are presented in the Supplementary Fig. 5, showing the capability and adaptability of the FND-receiver in managing complex modulation schemes in a multiple access system.

## Multi-band receiver and reference-free design

The multiple access system discussed above is not confined to operating near the zero-field splitting (ZFS) frequency but can be extended to other frequency bands, enabling multi-band communication. By applying an external static magnetic field $B$ along the NV axis, the ODMR spectrum comprises two Lorentzian peaks, as shown in Supplementary Fig. 1b (red curve). Each peak corresponds to a ground state spin transition at energy levels $D \pm \gamma B$, where $\gamma = 2.8$ MHz/G is the electron spin gyromagnetic ratio of the NV center. Such capability of modifying the frequency range of the peaks by adjusting the magnetic field facilitates the multi-band communication. This means that the system can be tuned to match the receiver's operating frequency, eliminating the need for distinct antennas as required by traditional radio receivers for different bands. This makes the proposed NV centers based receiver more versatile and efficient, as it can adapt to various frequency bands without the need for antenna changes.

To demonstrate the feasibility of a multi-band receiver, we examine the same communication scenario described in the aforementioned multiple access system. By analyzing the ODMR spectra of FNDs within an 80 μm field of view, we observe that when an external magnetic field is applied, the smallest Lorentzian peak in the ODMR spectra of most FNDs is approximately 2776 MHz. Figure 4a displays the ODMR spectrum of one of the FNDs (marked by a blue circle). In the frequency range from 2700 MHz to 2776 MHz, bits 0 and 1 are respectively mapped to frequencies 2700 MHz and 2776 MHz for digital frequency modulation. The remaining procedures are the same as those in the previously discussed multiple access case. The subsequently recovered image messages are presented in Fig. 4a, with a BER measured as 0.428%. The corresponding BEP, considering its uncertainty, is estimated to be (0.004 ± 0.001) at a 95% confidence level. The current frequency range employed corresponds to the ground state transition $|0\rangle \rightarrow |-1\rangle$ of NV centers, while the transition $|0\rangle \rightarrow |+1\rangle$ enables an extended operating

frequency range exceeding 2870 MHz to meet communication systems' requirement. Figure 4b demonstrates the capability of operating at higher frequency bands for transmission within the range from 2963 MHz to 3020 MHz, with the recovered image messages presented and a measured BER of 0.123%. The related BEP is specified as (0.0012 ± 0.0006) at a 95% confidence level.

Finally, we propose a reference-free design for orthogonal multiple access, addressing the issue of communication overhead caused by the transmission of reference bits. Due to the presence of different axis of randomly distributed FNDs and the existence of magnetic field gradients when an external magnetic field is applied, multiple FNDs can be observed within a certain field of view without their Lorentzian peaks overlapping in the ODMR spectrum. The left curves in Fig. 4c depict the ODMR spectra of two FNDs, labelled by blue and green circles, within an 18 μm field of view. The transitions $|0\rangle \rightarrow |+1\rangle$ in one kind of NV centers in two FNDs occur at frequencies of 2946.5 MHz and 2959.5 MHz, respectively. Within this field of view, applying a small static magnetic field leads to magnetic field gradients of around 0.023 G/um, resulting in an ODMR drift of approximately 1 MHz (Supplementary Fig. 7). Consequently, the discrepancy in their Lorentzian peaks primarily arises from the distinct axial orientations of the two FNDs. Orthogonal dual access is achieved by incorporating the two distinct FNDs into the receiver's cover glass, with both links functioning as parallel connections. The separation between the two FNDs is merely around 10 μm, much smaller than the half-wavelength spacing (around 150 mm) typically required between antenna elements in radio receivers. For digital frequency modulation, bits 0 and 1 of one user map to frequencies 2946.5 MHz and 3000 MHz, respectively, while bits 0 and 1 of the other user map to 2959.5 MHz and 3000 MHz, respectively. The remaining transmission procedures are the same as those in the previously described multiple access scenario. The recovered image messages with a BER of 0.0657% are displayed on the right side of Fig. 4c, validating the feasibility and efficiency of this approach. The estimated BEP, factoring in its uncertainty, is given as (0.0006 ± 0.0004) at a 95% confidence level. Furthermore, although we have demonstrated the transmission for two users in this example, a broader field of view enables the identification of multiple appropriate FNDs. In particular, due to a restricted field of view and limited axial orientations, our FND-receiver in the experiment can accommodate up to 5 users within a 75 μm × 75 μm field of view and magnetic field gradients of approximately 0.023 G/μm, yielding a utilization ratio of 11.1% (5 out of 45) for FNDs. A comprehensive discussion on the number of channels can be found in Supplementary Note 5. The number of channels is determined by the frequency separation of Lorentzian peaks, which is affected by the external magnetic field gradients and field of view. Specifically, an extensive field of view allows for the automatic demultiplexing of signals from a considerable number of users without the need for transmitting reference bits. The scalability of the system to accommodate more users would come with several trade-offs in terms of power, space, and complexity as discussed in Supplementary Note 6.

## Implementation of a compact device

To demonstrate the feasibility and practicality of the proposed FND-receiver, we constructed a compact device containing all essential optical components, as illustrated in Fig. 5a, with fabrication details provided in Methods. A photograph of the assembled compact FND-receiver device is displayed in Fig. 5b. The details of the size, weight, and power consumption of our implemented receiver are presented in Methods and Supplementary Note 9. We evaluate the performance of the implemented device by conducting tests following procedures similar to those discussed in the previous section, focusing on the simultaneous detection of two transmitted signals from two distinct users. The experiments involve transmitting two black-and-white images, with the received images presented in Fig. 5c. In the experiment, the BERs are measured as 1.79% and 0.38% for the two received image messages, respectively, and the overall BER is calculated to be 1.08%. The estimated BEP, including its uncertainty, is provided as (0.011 ± 0.005) at a 95% confidence level. Although this value is higher than that obtained in

**Fig. 4 | Experimental demonstration of a multi-band receiver and reference-free design. a** Lower frequency band (2700 MHz to 2776 MHz) and **b** higher frequency band (2963 MHz to 3020 MHz). (Left) Optically detected magnetic resonance (ODMR) spectrum of the fluorescent nanodiamond (FND) marked by a blue circle. (Right) The recovered images at the FND-receiver's end. **c** Reference-free orthogonal multiple access. (Left) ODMR spectra of the two FNDs labelled by blue and green circles. (Right) The recovered images at the FND-receiver's end.

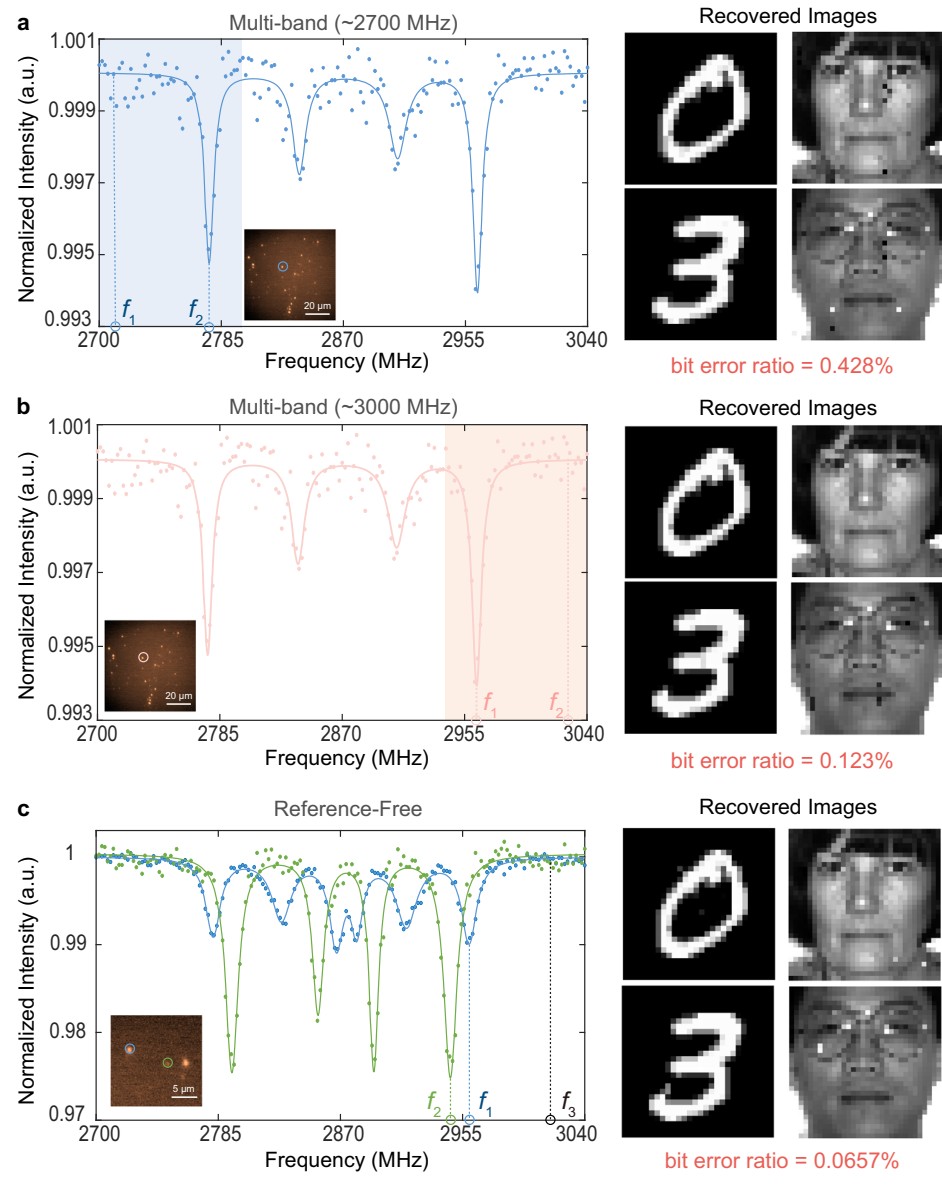

## Discussion

This work validates the feasibility of our proposed NV centers based quantum receivers for enabling ubiquitous radio access. Specifically, we design the FND-receiver to support multiple access, and the experimental results reveal its advantages, e.g., it induces low BER and requires low-complexity facilities. To substantiate our findings, we implemented a compact device as a demonstrative example for a proof of practical FND-receiver, with tests on this device exhibiting acceptable performance metrics in wireless communication systems.

Following that, we explored the factors influencing communication performance. First, we conducted experiments to examine the impact of the number of fluorescence spots, i.e., FND clusters, in the field of view. As shown in Fig. 5d, the BER generally decreases as the number of fluorescence spots increases. This observation aligns with the expectation that a greater number of FNDs provide more distinctive points for differentiating various signal combinations, which can be interpreted as a higher feature space dimension. Second, we performed experiments to assess the effect of laser power on performance. As illustrated in Fig. 5e, when the laser power is low,

experiments conducted using the laboratory precise facilities, it remains noteworthy given the uncoded transmission adopted in the experiments.

e.g., 5 mW, the BER is considerably high (~35%). This can be attributed to the reduced brightness of fluorescence from FNDs at lower laser power levels, which results in smaller intensity differences and less distinguishable responses to different bit pairs, leading to a higher BER. Moreover, the lower brightness of FNDs allows only a limited number of fluorescence spots to be detected by the CMOS sensor under this regime, thereby reducing the number of characteristic points. As the laser power increases, more fluorescence spots become visible, enhancing the feature dimension and enabling more accurate bits detection. The sharp decrease in BER can be ascribed to the improved discernibility and discrimination of FND clusters at higher laser power levels. Furthermore, we conducted additional experiments to evaluate the relationship between BER and signal strength (microwave power). The plot of BER as a function of signal strength is shown in Supplementary Fig. 10. From the results, it is evident that higher signal strength consistently leads to a lower BER. This is because stronger microwave signals enhance the SNR, improving the system's ability to distinguish between different symbols and reducing the likelihood of errors in symbol detection. Such ability is related to the sensitivity and ODMR contrast as discussed in Supplementary Note 4.

In summary, we posit that the development of the proposed FND-receiver leveraging NV centers into a multipurpose technology presents a

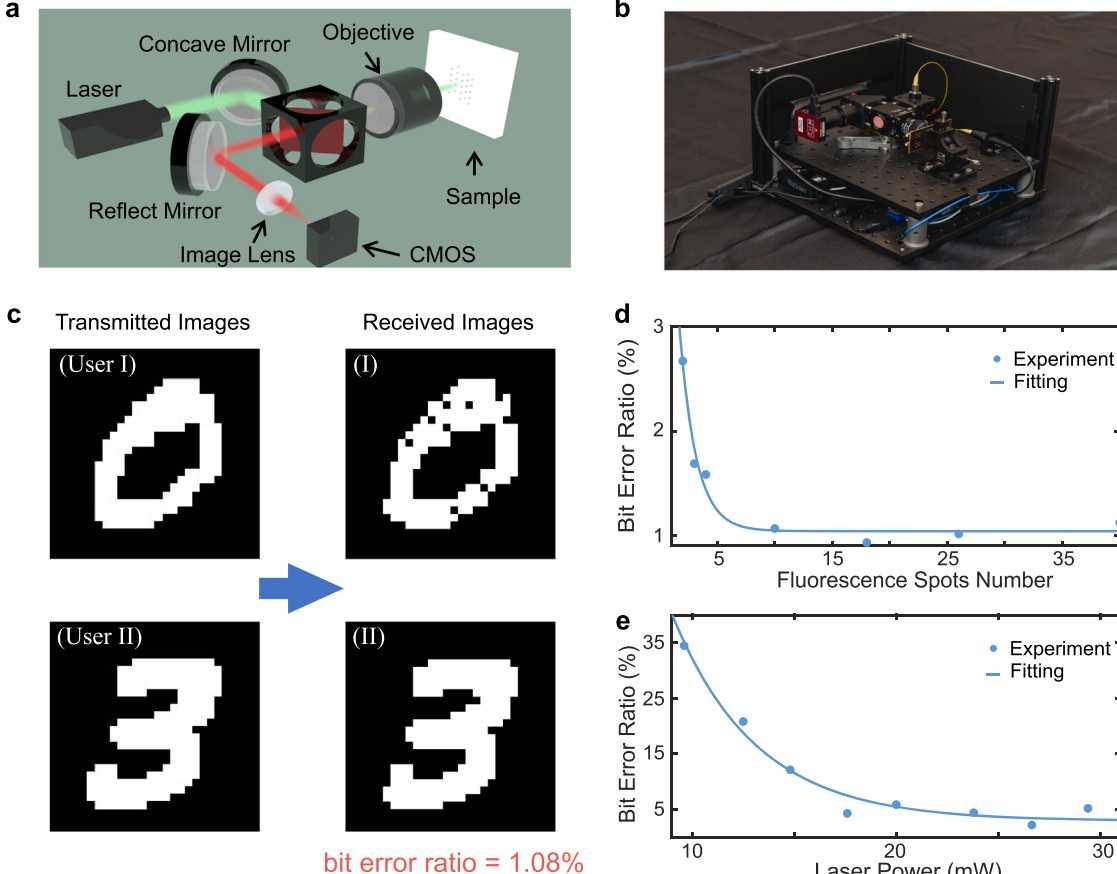

**Fig. 5 | Implementation and testing of a compact fluorescent nanodiamond (FND)-receiver. a** The schematic shows the components and overall structure of the implemented compact device. It consists of key elements such as a laser, a concave mirror, a dichroic mirror, a CMOS sensor, and an objective lens. **b** A photograph of the fabricated compact device. **c** Experiments involving the transmission of two black-and-white images: (Left) the transmitted images; (Right) the received images. **d, e** The impact of the number of fluorescence spots and laser power on the communication performance measured by the bit error ratio (BER).

feasible approach to actualizing the 6G vision, catering to future services and applications requiring ubiquitous radio access. The demonstrated benefits of miniaturization, low BER, and adaptability across various frequency bands make it a compelling receiver candidate for the next generation of wireless communication networks. Furthermore, an in-depth analysis of the practical requirements for 6G from various aspects such as frequency band, bandwidth, symbol rate, channel number, crosstalk, and minimum detected power is provided in Supplementary Note 8. Specific focus is given to bandwidth and symbol rate as follows. Typical magnetic fields in practical implementations can reach several Tesla, enabling the bandwidth of FND-receiver ranging from several GHz to hundreds of GHz. This allows the system to support millimeter-wave bands (30–300 GHz), which are envisioned as the working band for 6G networks[32]. Extending to higher frequency bands, such as terahertz bands, requires down-converting the target signals into the NV center's detection range using a frequency mixer. Recent studies have demonstrated the feasibility of such an approach, achieving broadband microwave detection using spins in diamond interfaced with a thin-film magnet[26]. On the other hand, constrained by the transition rate between energy levels[33], that is $0.98 \pm 0.31$ MHz, the potential symbol rate of FND-receiver can achieve 1 mega-symbols per second, which exceeds that of current traditional receivers (0.014-0.22 mega-symbols per second) according to the 5G NR standards (Supplementary Note 8).

## Methods
### Sample preparation
The coverslip substrate was initially treated to enhance its surface hydrophilicity using the UV-Ozone Cleaner equipment (UV250-MC) at a power of 1.3 W for 10 minutes. Subsequently, commercially available surface-

modified fluorescence nanodiamonds (FNDs) measuring 100 nm (FND Biotech, 100 nm Red FND-COOH, 1 mg/ml) were dissolved in milli-Q water and diluted to a concentration of 0.05 mg/ml. A volume of 300 ul of the diluted FND solution was subjected to sonication for 5 minutes. Then, 20 ul of the sonicated solution was carefully dropped onto the coverslip substrate. The substrate was then dried using a spin coater (KW-4A) at a low rotating speed of 1500 r/min for 20 seconds, followed by a high speed of 3000 r/min for 50 seconds. This process was repeated 10 times to ensure complete spin coating of the FND onto the substrate's surface.

### Experimental setup
The NV photoluminescence image was acquired using a home-built upright widefield microscope. Excitation of FNDs was achieved using a green laser with a wavelength of 532 nm. The emitted fluorescence was collected by an Electron-Multiplying CCD (EMCCD) camera (Evolve 512 Delta, Photometrics) equipped with a long-pass filter of 650 nm. To facilitate the experiment, the FND substrate was placed on a self-designed PCB board with a microwave antenna. The two signal microwaves required for the experiment were generated by a Windfreak Technologies microwave source (SynthHD), controlled by a Mini-Circuits RF switch (ZASWA-2-50DRA), and amplified using a Mini-Circuits amplifier (ZHL-16W-43-S + ). The synchronization of microwave transmission and image acquisition was achieved using a pulse generator (Pulse streamer, Swabian). All measurements were performed at room temperature.

### Compact device fabrication
To achieve a compact design, a Pigtailed Laser Diode (LP520-SF15A) was used in the excitation light path. The laser beam was collimated using a fiber

collimator (F220FC-532) to produce parallel light with a spot diameter of 2.1 mm. A concave mirror (CM254-100-E02) with a focal length of 100 mm was employed in place of a convex lens of the same focal length to reduce space requirements. For fluorescence collection, a CMOS camera (Basler CMOS, aca4024-29um) was used, which helped in saving space and reducing costs. The objective lens used (Olympus UPlanSApo 40X/0.95NA) was the same as the one used in the home-built microscope.

## Size, weight, and power consumption implemented receiver

Our current receiver prototype measures 300 mm × 300 mm × 200 mm, encompassing all optical and electronic components. The total weight is approximately 8 kg, with optical breadboards contributing 6 kg and the remaining optical and electronic components weighing below 2 kg. In our receiver prototype, the power consumption for the camera and laser are 1.17 W and 0.7 W respectively, resulting in a total power consumption of 1.87 W. Details about the improvements that can be made to the receiver in terms of its size, weight, and power consumption, as well as an exploration of its potentials and limitations, can be found in Supplementary Note 9.

## NV microwave transmission measurement

The entire measurement process was conducted using continuous wave optically detected magnetic resonance (CWODMR) technique. Each image acquisition time was set at 40 ms for the Electron-Multiplying CCD (EMCCD) camera. During the entire readout time, the microwave was applied for a duration of 30 ms, with an additional 10 ms allocated for temperature balancing. To ensure a high-quality signal, a total of 400 measurements were performed on the reference image. These measurements were then averaged to obtain the final image, effectively reducing any noise or fluctuations in the data. In contrast, the transmission data image was only measured once. For a more detailed pulse sequence, please refer to Supplementary Fig. 9.

## Data availability

The authors declare that the data supporting the findings of this study are available within the article and its Supplementary Information file. All other data that support the findings of the study are available from the corresponding authors upon request.

## Code availability

MATLAB (R2023b) codes used for processing data that support the findings of this study are available from the corresponding authors upon request.

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

## Acknowledgements

The work of Q.Z. and K.H. was supported in part by a Fellowship Award from the Hong Kong (HK) Research Grants Council (RGC) under Grant HKU RFS2122-7S04; in part by General Research Fund (GRF) from HK RGC under Grant 17212423; in part by Areas of Excellence (AoE) Scheme from HK RGC under Grant AoE/E-601/22-R; and in part by Collaborative Research Funding (CRF) Scheme from HK RGC under Grant C1009-22GF. Z.C. acknowledges the financial support from the National Natural Science Foundation of China (NSFC) and the Research Grants Council (RGC) of Hong Kong Joint Research Scheme (Project No. N_HKU750/23); HKU Seed Fund; and the Health@InnoHK program of the Innovation and Technology Commission of the Hong Kong SAR Government. The authors appreciate Mr. Linjie Ma from HKU EEE for photographing the compact device.

## Author contributions

Z.C. and K.H. initialized the idea and supervised the project. Q.Z. and J.Z. finalized the idea and arranged specific tasks. Q.Z. designed the techniques for supporting multiple access and processed the experimental data. J.Z. performed the experimental measurements and implemented the compact device. M.G. configured the laser and associated electronic components for the compact device. Q.Z. and J.Z. wrote the manuscript and supplementary information. All authors discussed the results and commented on the manuscript.

## Competing interests

The authors declare no competing interests.
