## [Transparent Peer Review file · Communications Engineering]

Towards ubiquitous radio access using nanodiamond based quantum receivers

Corresponding Author: Professor Kaibin Huang

Version 0:

Reviewer comments:

Reviewer #1

(Remarks to the Author)

The article presents a novel approach to multiple-access wireless communication using fluorescent nanodiamonds containing nitrogen-vacancy (NV) centers. The NV centers act as quantum receivers, where a microwave signal, through coupling to the NV center resonant transition, modulates the intensity of fluorescence excited by a green laser. Experimental demonstrations show that the system can upconvert modulated microwave signals to the optical domain, which is subsequently detected with a camera. Leveraging unique patterns from the NV center distribution, multiple users or channels can be accessed simultaneously by referencing individual NV distribution patterns. A low bit error ratio (BER) was demonstrated in a two-channel system, showcasing the system's ability to operate efficiently. The work highlights the potential of these quantum receivers to be miniaturized and adapted for next-generation communication networks. The concept is innovative, and the manuscript is well written with thorough details.

Before publication, the authors should address the following comments:

Comments:

- The modulation depth seems to be small, only a few percent. How viable is this for practical applications, and what specific requirements does this impose on the communication channel such as dynamic range and optical power needed?
- In your multi-user system, does the overhead associated with identifying each bit-pair scale linearly with the number of users or data points?
- What is the downside of using the reference-free scheme?
- For amplitude modulation, the BER is reported as 0%. Could this result from limited testing data or a small sample size? Would it be more precise to state that the BER is less than a threshold determined by the number of data points?
- The system's operation speed seems constrained by the camera acquisition time (40 ms). What factors limit this, and could increasing laser power or other improvements reduce the acquisition time?
- What are the current bandwidth and data rate limitations of this NV center-based receiver system? Can it meet the requirements of next-generation 6G networks, and how might it scale with future demand?
- The system currently operates at 2.8 GHz. With new wireless systems expected to operate in the millimeter or terahertz bands, can this NV center-based concept be extended to support those higher frequency bands?
- Can you clarify the differences in the implementation shown in Fig. 5 compared to previous demonstrations of NV center-based receivers? What improvements or modifications were made?
- The author claim the system is scalable to accommodate more users. What would be the trade-offs in terms of power, space, or complexity?
- How does the proposed NV center-based system for microwave-to-optical up-conversion compare to other platforms, for example, based on Silicon photonics (10.1109/JLT.2014.2319152), lithium niobate (10.1109/JSTQE.2013.2265192), plasmonics (<https://www.nature.com/articles/s41566-018-0281-6>), in terms of efficiency, bandwidth, and scalability? The authors should at least mention these other platforms and suggest advantages their platform bring.
- Line 41: change to "consisting"

Reviewer #3

(Remarks to the Author)

This work proposes to use nanodiamonds for multi-channel RF sensing, in particular focusing on their potential application as a microwave receiver. To the best of my knowledge this is the first demonstration of a multichannel nanodiamond based microwave receiver, offering an intriguing new detection technique. However I feel that the paper needs more quantitative

detail and additional rigour in their results, and in particular should more thoroughly consider whether the proposed system has the future potential for genuine utility. Overall I believe if the researchers strengthen their work with these details I think it is of sufficient interest to the field of engineering to merit publication in Comms Engineering. Below are my suggestions for improving this manuscript.

- How do the specifications of the implemented receiver compare to those required for a 6G network, what is the signal bandwidth, channel number, crosstalk, minimum detected power, etc for this system. The authors should more thoroughly discuss the measured and potential performance of such a system compared to the current state of the art.
- Are all BERs determined from just a single transfer of an image? What is the uncertainty in the bit error rates for all examples? They should be quoted. These numbers should be calculated by averaging over a sufficient quantity of received data to provide valid uncertainties in the error rates.
- What is the maximum bitrate of the receiver? What is the limitation? If the limitation is the CMOS camera, what is the limiting bandwidth of the NV sensors themselves, and what causes that limitation? How does this bandwidth compare to currently used receivers, and how well does it fit with the vision of utilising this receiver for 6G communications?
- What is the sensitivity of the receiver? A plot of BER vs signal strength would be very instructive for some or all use cases.
- Is the bit detection process implemented in real-time or on pre-collected data? It seems very resource inefficient to need to process a 12.2 megapixel image for each bit. What is the maximum bitrate this sort of process is capable of maintaining in real time with suitable/specialised hardware?
- It was not very clear to me why the number of channels was limited to a maximum of 5, or why the utilisation rate is limited to 5/45. What is the estimation that has occurred to reach 5 users? Perhaps I have missed this discussion or estimation in the paper, but I think it is worthy of at least brief discussion in the main text and detailed discussion in the supplementary.
- What is the size, weight, and power consumption of this whole receiver design? What can be done to improve these and what is the potential and limitations in SWaP-C for large scale deployment?

Some minor details:

Line 41, "traditional receivers consists of a series...", should read "traditional receivers that consist of a series..."
Line 42, marking should be making.

Reviewer #4

(Remarks to the Author)

"I co-reviewed this manuscript with one of the reviewers who provided the listed reports. This is part of the Communications Engineering initiative to facilitate training in peer review and to provide appropriate recognition for Early Career Researchers who co-review manuscripts."

Version 1:

Reviewer comments:

Reviewer #1

(Remarks to the Author)

The authors have adequately addressed all my comments.

Reviewer #3

(Remarks to the Author)

The authors have improved the manuscript that neatly combines quantum sensors for novel RF applications. I can recommend this article to be published without further adjustments.

Reviewer #4

(Remarks to the Author)

The authors have significantly improved the manuscript and I think it is a novel and interesting paper for the fields of quantum sensing, and RF communications. I can recommend this work for publication without any further peer review. I only have a couple of minor suggestions to improve the paper further:

Line 79 – "based on the such contrast" should read "based on this contrast"
Supplementary reference 10 is showing up as incomplete.

Supplementary Note 8, 6.) Minimum detected power, minimum detected power is quoted from the microwave source. This should be instead estimated and stated at the receiver based on the microwave amplification and transmission chain. A number that is equivalent to the reference signal received power (RSRP) number quoted in 5G comms should be presented as the minimum detected power. To me this is the number 1 concern for these type of receivers. Typical 5G receivers operate with received power in the -60 to -90dBm range. How many orders of magnitude away from this range is this receiver currently, and it should be stated how much each of the proposed improvements can achieve. A multi-access receiver still has utility even if its minimum received power for operation is higher than other single access alternatives but if there will be a trade-off it should be quantified.

Reviewers' comments:

Reviewer #1 (Remarks to the Author):

The article presents a novel approach to multiple-access wireless communication using fluorescent nanodiamonds containing nitrogen-vacancy (NV) centers. The NV centers act as quantum receivers, where a microwave signal, through coupling to the NV center resonant transition, modulates the intensity of fluorescence excited by a green laser. Experimental demonstrations show that the system can upconvert modulated microwave signals to the optical domain, which is subsequently detected with a camera. Leveraging unique patterns from the NV center distribution, multiple users or channels can be accessed simultaneously by referencing individual NV distribution patterns. A low bit error ratio (BER) was demonstrated in a two-channel system, showcasing the system's ability to operate efficiently. The work highlights the potential of these quantum receivers to be miniaturized and adapted for next-generation communication networks. The concept is innovative, and the manuscript is well written with thorough details.

Before publication, the authors should address the following comments:

Comments:

- The modulation depth seems to be small, only a few percent. How viable is this for practical applications, and what specific requirements does this impose on the communication channel such as dynamic range and optical power needed?
- In your multi-user system, does the overhead associated with identifying each bit-pair scale linearly with the number of users or data points?
- What is the downside of using the reference-free scheme?
- For amplitude modulation, the BER is reported as 0%. Could this result from limited testing data or a small sample size? Would it be more precise to state that the BER is less than a threshold determined by the number of data points?
- The system's operation speed seems constrained by the camera acquisition time (40 ms). What factors limit this, and could increasing laser power or other improvements reduce the acquisition time?
- What are the current bandwidth and data rate limitations of this NV center-based receiver system? Can it meet the requirements of next-generation 6G networks, and how might it scale with future demand?
- The system currently operates at 2.8 GHz. With new wireless systems expected to operate in the millimeter or terahertz bands, can this NV center-based concept be extended to support those higher frequency bands?
- Can you clarify the differences in the implementation shown in Fig. 5 compared to previous demonstrations of NV center-based receivers? What improvements or modifications were made?
- The author claim the system is scalable to accommodate more users. What would be the trade-offs in terms of power, space, or complexity?
- How does the proposed NV center-based system for microwave-to-optical up-conversion compare to other platforms, for example, based on Silicon photonics (10.1109/JLT.2014.2319152), lithium niobate (10.1109/JSTQE.2013.2265192), plasmonics (<https://www.nature.com/articles/s41566-018-0281-6>), in terms of efficiency, bandwidth, and scalability? The authors should at least mention these other platforms and suggest advantages their platform bring.
- Line 41: change to "consisting"

Reviewer #3 (Remarks to the Author):

This work proposes to use nanodiamonds for multi-channel RF sensing, in particular focusing on their potential application as a microwave receiver. To the best of my knowledge this is the first demonstration of a multichannel nanodiamond based microwave receiver, offering an intriguing new detection technique. However I feel that the paper needs more quantitative detail and additional rigour in their results, and in particular should more thoroughly consider whether the proposed system has the future potential for genuine utility. Overall I believe if the researchers strengthen their work with these details I think it is of sufficient interest to the field of engineering to merit publication in Comms Engineering. Below are my suggestions for improving this manuscript.

- How do the specifications of the implemented receiver compare to those required for a 6G network, what is the signal bandwidth, channel number, crosstalk, minimum detected power, etc for this system. The authors should more thoroughly discuss the measured and potential performance of such a system compared to the current state of the art.
- Are all BERs determined from just a single transfer of an image? What is the uncertainty in the bit error rates for all examples? They should be quoted. These numbers should be calculated by averaging over a sufficient quantity of received data to provide valid uncertainties in the error rates.
- What is the maximum bitrate of the receiver? What is the limitation? If the limitation is the CMOS camera, what is the limiting bandwidth of the NV sensors themselves, and what causes that limitation? How does this bandwidth compare to currently used receivers, and how well does it fit with the vision of utilising this receiver for 6G communications?
- What is the sensitivity of the receiver? A plot of BER vs signal strength would be very instructive for some or all use cases.
- Is the bit detection process implemented in real-time or on pre-collected data? It seems very resource inefficient to need to process a 12.2 megapixel image for each bit. What is the maximum bitrate this sort of process is capable of maintaining in real time with suitable/specialised hardware?
- It was not very clear to me why the number of channels was limited to a maximum of 5, or why the utilisation rate is limited to 5/45. What is the estimation that has occurred to reach 5 users? Perhaps I have missed this discussion or estimation in the paper, but I think it is worthy of at least brief discussion in the main text and detailed discussion in the supplementary.
- What is the size, weight, and power consumption of this whole receiver design? What can be done to improve these and what is the potential and limitations in SWaP-C for large scale deployment?

Some minor details:

Line 41, "traditional receivers consists of a series...", should read "traditional receivers that consist of a series..."

Line 42, marking should be making.

Reviewer #4 (Remarks to the Author):

“I co-reviewed this manuscript with one of the reviewers who provided the listed reports. This is part of the Communications Engineering initiative to facilitate training in peer review and to provide appropriate recognition for Early Career Researchers who co-review manuscripts.”

Reviewer #1

“The article presents a novel approach to multiple-access wireless communication using fluorescent nanodiamonds containing nitrogen-vacancy (NV) centers. The NV centers act as quantum receivers, where a microwave signal, through coupling to the NV center resonant transition, modulates the intensity of fluorescence excited by a green laser. Experimental demonstrations show that the system can upconvert modulated microwave signals to the optical domain, which is subsequently detected with a camera. Leveraging unique patterns from the NV center distribution, multiple users or channels can be accessed simultaneously by referencing individual NV distribution patterns. A low bit error ratio (BER) was demonstrated in a two-channel system, showcasing the system’s ability to operate efficiently. The work highlights the potential of these quantum receivers to be miniaturized and adapted for next-generation communication networks. The concept is innovative, and the manuscript is well written with thorough details. Before publication, the authors should address the following comments:”

Response:

We are grateful for the reviewer acknowledging the novelty of our work and have tried our best to address all the raised questions in detail in the subsequent point-to-point response.

Comment 1. *“The modulation depth seems to be small, only a few percent. How viable is this for practical applications, and what specific requirements does this impose on the communication channel such as dynamic range and optical power needed?”*

Response:

To address the comment, we would like to clarify this issue from the two aspects:

1) Modulation depth: The modulation depth, also referred to as optically detected magnetic resonance (ODMR) contrast, plays a crucial role in determining the sensitivity of the receiver. Particularly, the sensitivity is inversely proportional to the modulation depth, which means a higher modulation depth enhances the sensitivity and thus reduces the time required for signal acquisition in practical applications [R1]. The modulation depth is inherently limited by the physical properties of the nanodiamond itself. According to theoretical analysis and experimental results, a single nitrogen-vacancy (NV) center can exhibit a modulation depth (or contrast) of up to 30% [R2]. By comparison, a large ensemble of NV centers with random orientations typically shows a much smaller contrast of approximately 1% to 2% [R3]. This

reduction is primarily due to the averaging effect caused by the random alignment of NV centers. We note that the fluorescent nanodiamonds (FNDs) used in our work generally exhibit a contrast of around 10% near their zero-field splitting frequency, which has shown sufficient sensitivity for information signals detection and demultiplexing. Furthermore, the modulation depth can be improved through preferential NV alignment via material growth engineering [R4]. When NV centers are predominantly aligned along a single axis, the contrast can approach 30%, thereby improving sensitivity. We have added the description of contrast in the Manuscript page 3 (lines 78-81) and discussion in Supplementary Note 7 (page 20).

2) Practical applications and requirements: In practical applications, achieving higher modulation depth requires careful adjustment of laser power and microwave amplitude. The small modulation depth imposes specific requirements on the communication channel, particularly in terms of the dynamic range and optical power. i) Since higher microwave amplitudes tend to increase the contrast, a low modulation depth necessitates a relatively high SNR in transmission to reliably distinguish the small variations in the fluorescence signal associated with bit-state transitions. ii) Laser power has a more complex effect. The modulation depth initially increases with laser power due to enhanced optical pumping, but it decreases when optical pumping surpasses microwave driving at higher laser powers [R1]. Notably, increased laser power also enhances NV fluorescence, requiring a balance between maximizing fluorescence and optimizing ODMR contrast. Therefore, precise measurement of laser power is necessary to determine the optimal value for achieving the best balance of modulation depth and SNR. We have added a comprehensive discussion of modulation depth and its implications for dynamic range and laser power in the Supplementary Note 4 (pages 15-17).

Response Fig. 1. The effect of (a) laser power and (b) microwave power on the contrast of

ODMR. The black and blue curves are the plot of the right and left peaks of the ODMR spectrum, respectively.

References:

[R1] A. Dréau, M. Lesik, L. Rondin, P. Spinicelli, O. Arcizet, J-F. Roch, and V. Jacques. "Avoiding power broadening in optically detected magnetic resonance of single NV defects for enhanced dc magnetic field sensitivity." *Physical Review B—Condensed Matter and Materials Physics* 84, no. 19 (2011): 195204.

[R2] F. Jelezko, and J. Wrachtrup. "Single defect centres in diamond: A review." *Physica Status Solidi (a)* 203, no. 13 (2006): 3207-3225.

[R3] A. FL Poulsen, J. D. Clement, J. L. Webb, R. H. Jensen, L. Troise, K. Berg-Sørensen, A. Huck, and U. L. Andersen. "Optimal control of a nitrogen-vacancy spin ensemble in diamond for sensing in the pulsed domain." *Physical Review B* 106, no. 1 (2022): 014202.

[R4] C. Osterkamp, M. Mangold, J. Lang, P. Balasubramanian, T. Teraji, B. Naydenov, and F. Jelezko. "Engineering preferentially-aligned nitrogen-vacancy centre ensembles in CVD grown diamond." *Scientific Reports* 9, no. 1 (2019): 5786.

Comment 2. “*In your multi-user system, does the overhead associated with identifying each bit-pair scale linearly with the number of users or data points?*”

Response:

We would like to clarify this issue from two perspectives:

1) Reference bits and their overhead: The reference bits are designed to distinguish the fluorescence patterns generated by all possible bit-pair combinations. As a result, the overhead of reference bits increases exponentially, scaling as 2^k when there are k users in the system for multiple access. This approach remains practical if the number of users served by a single base station is relatively small, particularly in scenarios involving ubiquitous radio access where coverage cells are typically small.

2) Motivation for a reference-free scheme: This scalability challenge motivated us to develop the reference-free scheme, which eliminates the need for reference bits. By removing the reference bits, this scheme significantly reduces overhead, enabling efficient large-scale multiple access.

Comment 3. “*What is the downside of using the reference-free scheme?*”

Response:

The downside of using the reference-free scheme can be summarized in the following two aspects:

1) Increased hardware complexity and cost: The reference-free scheme requires the application of a static magnetic field to enable the differentiation of FNDs based on their ODMR measurements. This static magnetic field is used for creating a magnetic field gradient, which ensures that each FND exhibits a unique resonance frequency in the ODMR spectrum. However, implementing this requires additional hardware, such as precision-controlled magnets or magnet arrays, to generate a stable magnetic field across the field of view. This added complexity not only increases the design and manufacturing challenges but also raises the overall cost of the system.

2) Limited utilization of FNDs for multiplexing: The scheme necessitates pre-determining the ODMR of FNDs for selection. Due to the random orientations of the FNDs and the presence of magnetic field gradients when the external magnetic field is applied, multiple FNDs within a given field of view may exhibit distinct Lorentzian peaks in the ODMR spectrum. While this ensures that their peaks do not overlap, it also limits the number of FNDs that can be effectively utilized for multiplexing. For example, in our experiments, the FND-receiver was able to accommodate up to 5 users within a $75\ \mu\text{m} \times 75\ \mu\text{m}$ field of view with a magnetic field gradient of approximately $0.023\ \text{G}/\mu\text{m}$. This resulted in a utilization ratio of 11.1% (5 out of 45 FNDs), highlighting the constraints on the usage of FNDs.

Comment 4. “*For amplitude modulation, the BER is reported as 0%. Could this result from limited testing data or a small sample size? Would it be more precise to state that the BER is less than a threshold determined by the number of data points?*”

Response:

Yes, the reported BER of 0% is based on the results observed in our experiments. Please kindly note that our BER refers to *bit error ratio*, which is defined as the number of bit errors divided by the total number of transmitted bits during a studied time interval. It is an empirical value obtained by counting the errors in a known number of bits transmitted over the system. In general, it can be considered as an approximate estimate of the *bit error probability* (BEP)

which is a theoretical value that predicts the likelihood of a bit error occurring. For BEP, this outcome arises from the limited testing data, as the statistic is derived from the transmission of 14,464 bits. In this context, it is more precise to state that the BEP is less than a threshold determined by the sample size. Specifically, for our case, it is less than $1/14464$ which corresponds to approximately 0.0069%. We have mentioned this value in the Manuscript page 3 (lines 60-61) and page 5 (lines 139-140).

Comment 5. *“The system’s operation speed seems constrained by the camera acquisition time (40 ms). What factors limit this, and could increasing laser power or other improvements reduce the acquisition time?”*

Response:

Currently, our system operates with a total acquisition time of 40 ms, which includes 30 ms for camera acquisition and an additional 10 ms to trigger the microwave source. The primary factors limiting this acquisition time are listed as follows:

1) Microwave source limitations: The microwave source used in our setup (Windfreak Technology Synth HD) is unable to trigger rapidly enough to match the desired speed for changing microwave frequency or amplitude. Specifically, it requires over 5 ms to adjust the microwave parameters, even under a trigger pulse. This delay contributes to the overall acquisition time. However, this aspect pertains to the transmitter and is not a factor in our receiver’s design, even though it constrains the demonstration performance.

2) Camera speed limitations: The camera used in our experiments (Teledyne Photometrics Evolve 512 Delta) is limited to a maximum frame rate of 67 frames per second, corresponding to approximately 15 ms acquisition time per frame for a full-frame pixel readout. While we could theoretically reduce the acquisition time to around 20 ms (15 ms for the camera and 5 ms for the microwave source), this improvement would be insufficient using our current facility.

Several potential improvements could address these constraints and reduce the acquisition time:

1) Faster microwave sources: Replacing the current microwave sources with transmitters designed for commercial wireless communications could significantly reduce the delay associated with changing microwave parameters. These transmitters can achieve much faster triggering times, aligning better with high-speed requirements.

2) High-speed cameras: Utilizing event cameras, which are capable of high-speed ODMR imaging, could dramatically improve data acquisition rates. For example, event cameras have

been shown to achieve image acquisition rates of up to 10 kHz [R5]. This technology would enable much faster frame rates compared to our current camera.

3) Integration with photodetectors: Another approach involves integrating nanodiamonds on-chip [R6] and collecting fluorescence signals using photodetectors such as photodiodes, avalanche photodiodes, or photomultiplier tubes. These detectors can achieve acquisition rates as high as 100 kHz [R7, R8, R9]. This approach would bypass the need for slower camera-based systems, enabling real-time signal acquisition at much higher speeds.

We have added the discussions on this issue in Supplementary Note 7 (pages 20-21).

References:

[R5] Z. Du, M. Gupta, F. Xu, K. Zhang, J. Zhang, Y. Zhou, Y. Liu et al. "Widefield Diamond Quantum Sensing with Neuromorphic Vision Sensors." *Advanced Science* 11, no. 2 (2024): 2304355.

[R6] H. Siampour, S. Kumar, V. A. Davydov, L. F. Kulikova, V. N. Agafonov, and S. I. Bozhevolnyi. "On-chip excitation of single germanium vacancies in nanodiamonds embedded in plasmonic waveguides." *Light: Science & Applications* 7, no. 1 (2018): 61.

[R7] D. Le Sage, L. My Pham, N. Bar-Gill, C. Belthangady, M. D. Lukin, A. Yacoby, and R. L. Walsworth. "Efficient photon detection from color centers in a diamond optical waveguide." *Physical Review B—Condensed Matter and Materials Physics* 85, no. 12 (2012): 121202.

[R8] Q. Gu, L. Shanahan, J. W. Hart, S. Belser, N. Shofer, M. Atature, and H. S. Knowles. "Simultaneous nanorheometry and nanothermometry using intracellular diamond quantum sensors." *ACS Nano* 17, no. 20 (2023): 20034-20042.

[R9] J. Yun, K. Kim, S. Park, and D. Kim. "Temperature Selective Thermometry with Sub-Microsecond Time Resolution Using Dressed-Spin States in Diamond." *Advanced Quantum Technologies* 4, no. 11 (2021): 2100084.

Comment 6. "What are the current bandwidth and data rate limitations of this NV center-based receiver system? Can it meet the requirements of next-generation 6G networks, and how might it scale with future demand?"

Response:

To address your comment, we would like to clarify it as follows:

1) Bandwidth limitations: For the reference-based approach, the current bandwidth is determined by the linewidth of the ODMR, which is approximately 6–7 MHz. This represents an inherent limitation of the NV center’s physical properties. For the reference-free approach, the bandwidth is determined by the applied external static magnetic field, which can be calculated using the formula: $\Gamma = \gamma B$, where $\gamma = 28$ GHz/T is the NV center gyromagnetic ratio and B refers to the magnetic field component parallel to the NV axis. In practical systems, magnetic fields can easily reach several Tesla, resulting in bandwidths ranging from tens of GHz to hundreds of GHz. This large bandwidth is sufficient to meet the requirements of next-generation 6G networks in mid-band spectrum [R10]. However, if future demands were to require bandwidths in the THz range, such requirements would exceed the capabilities of the NV center-based receiver system. In such cases, alternative quantum systems with broader bandwidths could be explored. While our current study focuses on NV centers, we believe our approach demonstrates the feasibility of integrating general quantum systems into wireless communication systems, paving the way for further advancements in this area.

2) Data rate limitations: The current data rate in our experiments is constrained by hardware limitations, specifically the speed of the microwave source and the camera, resulting in a minimum acquisition time of approximately 20 ms. Once these hardware constraints are addressed, the data rate will instead be limited by the interaction time between the microwave field and the spin states of NV centers in the receiver. For continuous-wave (CW) ODMR, the NV spin polarization time requires sufficient microwave application time to observe contrast in fluorescence. This is constrained by the transition rate between energy levels. Specifically, it refers to the k_{s0} parameter shown in Response Fig. 2, with a limited rate of 0.98 ± 0.31 MHz [R11]. This rate determines the bit rate, implying that the acquisition rate will be limited to approximately a maximum of 1 mega-symbols per second in a solid-state quantum system. Achieving such speeds would require faster detectors, such as high-speed cameras or single-shot detectors like photodiodes, avalanche photodiodes, or photomultiplier tubes. In addition, achieving high data rates depends on the quality of the nanodiamonds. High-quality nanodiamonds with a large number of NV centers are critical to generating sufficiently bright fluorescence signals. With adequate brightness, the effective SNR would increase, achieving the shortest acquisition time (around 1 μ s) and the highest symbol rate.

Response Fig. 2. A diagram of energy levels with detailed transition paths.

3) Scalability and suitability for 6G networks: We would like to address this issue from two perspectives:

i) Since 6G is still under development and lacks standardization, we can compare our system to the current 5G standard. According to the 5G New Radio (NR) standards outlined in 3GPP [R12], a radio frame is fixed at 10 ms and consists of 10 subframes, each lasting 1 ms. Each subframe contains 1–16 slots, with each slot comprising 14 symbols. This results in a symbol duration ranging from 4.46 to 71.43 μ s, corresponding to a symbol rate of 0.014 to 0.22 mega-symbols per second. Notably, the potential symbol rate of FND-receiver (1 mega-symbols per second) exceeds that of current traditional receivers (0.014-0.22 mega-symbols per second) based on the 5G NR standards [R12].

ii) We would also clarify that different 6G application scenarios have distinct requirements in terms of latency, data rate, bandwidth, etc. Our system is specifically designed to target the scenario of ubiquitous radio access in the 6G networks. In comparison to state-of-the-art communication systems, our NV center-based receiver system shows significant promise for integration into 6G networks. Traditional receiver designs are limited in size due to the requisite half-wavelength spacing between antenna elements, and restricted bandwidth which necessitates multiple RF receivers for multi-band signals detection. Our design addresses these issues by offering the key advantages: NV centers operate in the quantum domain and exhibit greater sensitivity than traditional metal antennas, almost have **no spacing constraints**. This makes them promising candidates for developing compact and sensitive receivers. Moreover, in contrast to traditional receivers with components like RF antennas, analog filters, and mixers, NV-based receivers or sensors mainly use optical components for detection and down-

conversion, marking them **inherently immune to electrical noise**. In summary, the key advantages include: *i) Wide bandwidth*: The system can naturally support millimeter-wave bands and can be extended into terahertz frequencies using a frequency mixer. *ii) Quantum-enhanced properties*: NV centers offer unique advantages, such as high sensitivity and immunity to electrical noise, which could provide an edge over classical systems in specific applications.

We have added the discussion on the band limitation and suitability for 6G communications in the Supplementary Note 8 and data rate limitation in the Supplementary Notes 7 and 8 (pages 20-24).

References:

[R10] Qualcomm Whitepaper, “Vision, market drivers, and research directions on the path to 6G,” Dec. 2022. [Online] <https://www.qualcomm.com/content/dam/qcomm-martech/dm-assets/documents/Qualcomm-Whitepaper-Vision-market-drivers-and-research-directions-on-the-path-to-6G.pdf>

[R11] J. Klatzow, J. N. Becker, P. M. Ledingham, C. Weinzetl, K. T. Kaczmarek, D. J. Saunders, J. Nunn, I. A. Walmsley, R. Uzdin, and E. Poem. "Experimental demonstration of quantum effects in the operation of microscopic heat engines." *Physical Review Letters* 122, no. 11 (2019): 110601.

[R12] 3GPP TS 38.211 – Table 4.3.2-1.

Comment 7. “*The system currently operates at 2.8 GHz. With new wireless systems expected to operate in the millimeter or terahertz bands, can this NV center-based concept be extended to support those higher frequency bands?*”

Response:

In the context of the reference-free configuration, the operating frequency band (or bandwidth) is primarily determined by the magnetic field. It can be computed using the formula $\Gamma = \gamma B$, where where $\gamma = 28 \text{ GHz/T}$ is the NV center gyromagnetic ratio and B refers to the magnetic field component parallel to the NV axis. Given that present-day magnetic fields can readily attain several Tesla, the resulting bandwidths can span from tens to hundreds of GHz. Hence, we are confident that our NV center-based receiver system can be adapted to accommodate the millimeter-wave bands. In terms of the other configuration incorporating reference bits, we can employ a **frequency mixer** to transmute the received microwave signals, specifically those

within the millimeter or terahertz bands, to the sensor-spin frequency of 2.87 GHz. A recent publication has demonstrated the practicality of this methodology, achieving extensive microwave detection by leveraging spins in diamond interfaced with a thin-film magnet [R13]. Therefore, we are optimistic that this strategy can effectively extend our system's compatibility with higher frequency bands.

We have added the discussion on the frequency band in the Supplementary Note 8 (page 22).

Reference:

[R13] J. J. Carmiggelt, I. Bertelli, R. W. Mulder, A. Teepe, M. Elyasi, B. G. Simon, G. EW Bauer, Y. M. Blanter, and T. van der Sar. "Broadband microwave detection using electron spins in a hybrid diamond-magnet sensor chip." *Nature Communications* 14, no. 1 (2023): 490.

Comment 8. *“Can you clarify the differences in the implementation shown in Fig. 5 compared to previous demonstrations of NV center-based receivers? What improvements or modifications were made?”*

Response:

The key differences between the implementation shown in Fig. 5 and the demonstrations in Figs. 3 and 4 lie in the experimental setup and the design approach:

- The experiments in Figs. 3 and 4 were conducted in a laboratory setting using a complex and large-scale setup. These demonstrations focused on highlighting the fundamental principles and performance of NV center-based receivers under controlled conditions.
- In contrast, Fig. 5 presents a more compact design and prototype fabrication, validating its potential for miniaturization and practical implementation. This demonstrates progress toward meeting the requirements for ubiquitous applications, where smaller and more integrated systems are essential.

Comment 9. *“The author claim the system is scalable to accommodate more users. What would be the trade-offs in terms of power, space, or complexity?”*

Response:

The scalability of the system to accommodate more users would come with several trade-offs in terms of power, space, and complexity:

1) Power trade-offs: *i) Increased magnetic field strength:* To scale the system, a stronger magnetic field gradient may be required to create larger frequency separations between FNDs. This will increase power consumption. *ii) Higher optical power:* Serving more users would require detecting signals from more FNDs, necessitating higher optical power for excitation and detection, increasing energy demands.

2) Space trade-offs: *i) Larger field of view:* Accommodating more FNDs requires a larger field of view, which may increase the size of the receiver's optical and hardware components, consuming more physical space. *ii) FND density:* To fit more FNDs within the same field of view, their spatial separation must decrease, potentially reducing the distinctiveness of their resonance peaks if the applied external magnetic gradient remains unchanged.

3) Complexity trade-offs: *i) Signal demultiplexing:* As more users are added, the system must resolve more closely spaced ODMR peaks, increasing the complexity of signal processing. *ii) Alignment challenges:* With more FNDs, precise alignment and calibration of the magnetic field and optical system become more difficult.

We have added the discussion on the tradeoffs in the Manuscript page 7 and Supplementary Note 6 (page 19).

Comment 10. “*How does the proposed NV center-based system for microwave-to-optical up-conversion compare to other platforms, for example, based on Silicon photonics (10.1109/JLT.2014.2319152), lithium niobate (10.1109/JSTQE.2013.2265192), plasmonics (<https://www.nature.com/articles/s41566-018-0281-6>), in terms of efficiency, bandwidth, and scalability? The authors should at least mention these other platforms and suggest advantages their platform bring.*”

Response:

The main advantages of an NV center-based system over electro-optical systems [R14, R15, R16] include low-loss transmission, lower laser power requirements, and high sensitivity for weak signal recovery.

1) Low-loss transmission: Electro-optical systems experience loss when interacting with microwave signals during laser detection, whereas our system utilizes fluorescence for transmitting microwave signals, resulting in minimal loss.

2) Lower laser power: Electro-optical systems typically require high laser power levels for a sufficient SNR, with the power requirement exceeding 10 mW for chip designs. In contrast, an

NV center-based system, particularly when employing a reference-free scheme and integrating FNDs on-chip [R17], can operate efficiently with laser power levels around 100 μ W.

3) High sensitivity for weak signal recovery: The NV center-based systems exhibit high sensitivity for weak signal recovery, with the ability to detect microwave magnetic fields as low as μ T, showcasing superior performance compared to electro-optical systems in terms of dynamic range for microwave signals [R18].

We have added a brief discussion on the comparison with the mentioned references in the introduction section of the Manuscript page 2 (lines 38-40).

References:

[R14] X. Zhang, A. Hosseini, H. Subbaraman, S. Wang, Q. Zhan, J. Luo, A. K-Y. Jen, and R. T. Chen. "Integrated photonic electromagnetic field sensor based on broadband bowtie antenna coupled silicon organic hybrid modulator." *Journal of Lightwave Technology* 32, no. 20 (2014): 3774-3784.

[R15] Y. N. Wijayanto, H. Murata, and Y. Okamura. "Electrooptic millimeter-wave–lightwave signal converters suspended to gap-embedded patch antennas on low- k dielectric materials." *IEEE Journal of Selected Topics in Quantum Electronics* 19, no. 6 (2013): 33-41.

[R16] Y. Salamin, B. Bäuerle, W. Heni, F. C. Abrecht, A. Josten, Y. Fedoryshyn, C. Haffner, R. Bonjour, T. Watanabe, M. Burla, D. L. Elder, L. R. Dalton, J. Leuthold. "Microwave plasmonic mixer in a transparent fibre–wireless link." *Nature photonics* 12, no. 12 (2018): 749-753.

[R17] H. Siampour, S. Kumar, V. A. Davydov, L. F. Kulikova, V. N. Agafonov, and S. I. Bozhevolnyi. "On-chip excitation of single germanium vacancies in nanodiamonds embedded in plasmonic waveguides." *Light: Science & Applications* 7, no. 1 (2018): 61.

[R18] Z. Wang, et al. "Picotesla magnetometry of microwave fields with diamond sensors." *Science Advances* 8.31 (2022): eabq8158.

Comment 11. "Line 41: change to "consisting""

Response:

We sincerely thank you for your careful reading. We have corrected the typo as you pointed out. The whole manuscript has been carefully proofread as best as we can.

Reviewer #3

“This work proposes to use nanodiamonds for multi-channel RF sensing, in particular focusing on their potential application as a microwave receiver. To the best of my knowledge this is the first demonstration of a multichannel nanodiamond based microwave receiver, offering an intriguing new detection technique. However I feel that the paper needs more quantitative detail and additional rigour in their results, and in particular should more thoroughly consider whether the proposed system has the future potential for genuine utility. Overall I believe if the researchers strengthen their work with these details I think it is of sufficient interest to the field of engineering to merit publication in Comms Engineering. Below are my suggestions for improving this manuscript.”

Response:

We appreciate the reviewer’s effort and constructive comments our paper. All your concerns are addressed in detail in the subsequent point-to-point response.

Comment 1. *“How do the specifications of the implemented receiver compare to those required for a 6G network, what is the signal bandwidth, channel number, crosstalk, minimum detected power, etc for this system. The authors should more thoroughly discuss the measured and potential performance of such a system compared to the current state of the art.”*

Response:

To address your comment, we provide a detailed explanation of the measured and potential performance of our system in terms of bandwidth, channel number, crosstalk, and minimum detectable power, and compare these characteristics to the requirements of next-generation 6G networks.

1) Bandwidth: For the reference-free scheme, the operating frequency band (or bandwidth) is determined by the applied magnetic field and can be calculated using the formula: $\Gamma = \gamma B$, where $\gamma = 28$ GHz/T is the nitrogen vacancy (NV) center gyromagnetic ratio and B refers to the magnetic field component parallel to the NV axis. Typical magnetic fields in practical implementations can easily reach several Tesla, resulting in bandwidths ranging from several GHz to hundreds of GHz. This capability positions the NV center-based system to support millimeter-wave bands (30–300 GHz), which are envisioned as the working band for 6G networks [R1]. For the reference-based scheme, the NV center operates at its intrinsic spin resonance frequency of 2.87 GHz. To extend the system to higher frequency bands, such as

terahertz bands, a frequency mixer can be utilized to down-convert the target signals into the NV center's detection range. Recent studies have demonstrated the feasibility of such an approach, achieving broadband microwave detection using spins in diamond interfaced with a thin-film magnet [R2]. This method allows the system to support higher frequency bands, potentially extending into the terahertz range (0.1–10 THz) depending on the mixer's performance and design.

2) Channel number: Channel number is a critical parameter for multi-user communication in 6G networks. In our current setup, for the reference-free scheme, the channel number is determined by the number of distinct microwave frequencies that can be addressed simultaneously without overlap. Here, the magnetic field gradient creates a frequency separation between NV centers across the field of view, allowing multiple channels to coexist. The number of channels is limited by the available bandwidth provided by the magnetic field (up to hundreds of GHz) and the linewidth of the optically detected magnetic resonance (ODMR) signal (approximately 6–7 MHz). In our experiment, 5 out of 45 fluorescent nanodiamonds (FNDs) can be distinguished, resulting in a utilization ratio of 11.1%. Thus, when we broaden the field of view to thousands of FNDs, we can accommodate up to hundreds of users which is sufficient for ubiquitous radio access in 6G network. For the reference-based scheme, the channel number could be increased by employing multiplexing techniques, such as spatial-division multiplexing by spatially separating NV centers using patterned fields or multiple spatially resolved detectors. Future work could focus on enhancing channel number by optimizing the magnetic field gradient and the NV center alignment.

3) Crosstalk: Crosstalk occurs when signals from adjacent channels interfere with one another, degrading the system performance. In our NV center-based system, crosstalk is primarily influenced by the linewidth of the ODMR signal and the precision of frequency separation. For the reference-free scheme, the use of a magnetic field gradient reduces crosstalk by separating NV center resonance frequencies spatially and spectrally. In the reference-based scheme, employing a high-quality narrowband filtering can minimize interference between channels. Additionally, the quantum nature of NV centers enables highly precise signal processing, which inherently reduces crosstalk compared to classical systems.

4) Minimum detected power: The dynamic range of the microwave power in our system spans approximately 30 dBm. The minimum microwave power detectable by the system is -25 dBm at the microwave source. The 6G systems are expected to operate with extremely low power levels, particularly for IoT devices and energy-efficient applications. While our current system

demonstrates a reasonable dynamic range, further improvements to sensitivity may be required to meet the stringent power efficiency standards of 6G. Enhancing the SNR through high-quality NV centers, optimized microwave amplifiers, and advanced detection techniques (e.g., lock-in detection) could lower the minimum detected power threshold.

5) Comparison to 6G requirements and future potential: In comparison to state-of-the-art communication systems, our NV center-based receiver system shows significant promise for integration into 6G networks. Traditional receiver designs are limited in size due to the requisite half-wavelength spacing between antenna elements, and restricted bandwidth which necessitates multiple RF receivers for multi-band signals detection. Our design addresses these issues by offering the key advantages: NV centers operate in the quantum domain and exhibit greater sensitivity than traditional metal antennas, have no spacing constraints. This makes them promising candidates for developing compact and sensitive receivers. Moreover, in contrast to traditional receivers with components like RF antennas, analog filters, and mixers, NV-based receivers or sensors mainly use optical components for detection and down-conversion, marking them inherently immune to electrical noise. In summary, the key advantages include: *i) Wide bandwidth:* The system can naturally support millimeter-wave bands and can be extended into terahertz frequencies using a frequency mixer. *ii) Quantum properties:* NV centers offer unique advantages, such as high sensitivity and low noise, which could provide an edge over classical systems in specific applications.

We have added a detailed discussion of these performance metrics and their potential for meeting 6G requirements to the Supplementary Note 8 (pages 22-24).

Reference:

[R1] W. Hong, Z.-H. Jiang, C. Yu, D. Hou, H. Wang, C. Guo, Y. Hu, L. Kuai, Y. Yu, Z. Jiang, Z. Chen. "The role of millimeter-wave technologies in 5G/6G wireless communications." *IEEE Journal of Microwaves* 1, no. 1 (2021): 101-122.

[R2] J. J. Carmiggelt, I. Bertelli, R. W. Mulder, A. Teepe, M. Elyasi, B. G. Simon, G. EW Bauer, Y. M. Blanter, and T. van der Sar. "Broadband microwave detection using electron spins in a hybrid diamond-magnet sensor chip." *Nature Communications* 14, no. 1 (2023): 490.

Comment 2. "Are all BERs determined from just a single transfer of an image? What is the uncertainty in the bit error rates for all examples? They should be quoted. These numbers

should be calculated by averaging over a sufficient quantity of received data to provide valid uncertainties in the error rates.”

Response:

Please kindly note that our BER refers to *bit error ratio*, which is defined as the number of bit errors divided by the total number of transmitted bits during a studied time interval. It is an empirical value obtained by counting the errors in a known number of bits transmitted over the system. In general, it can be considered as an approximate estimate of the *bit error probability* (BEP) which is a theoretical value that predicts the likelihood of a bit error occurring. For the BEP, this result arises from the limited testing data, as the statistic is derived from the transmission of 14,464 bits. Take the frequency modulation in Fig. 3 as an example, the uncertainties in BEPs are provided as follows:

The total number of transmitted bits is $n = 14,464$, among which $n_e = 45$ bits occur errors. Using the normal approximation (de Moivre-Laplace theorem), the BEP is estimated from the observations as $p = \frac{n_e}{n} \pm \frac{z_\alpha}{\sqrt{n}} \sqrt{\frac{n_e(n-n_e)}{n^2}}$, where z_α is the $1 - \frac{\alpha}{2}$ quantile of a standard normal distribution corresponding to the target error rate α . For a 95% confidential level, the error $\alpha = 0.05$ and $z_\alpha = 1.96$. Thus, BEP with its uncertainty is calculated as (0.0031 ± 0.0009) .

Following the same procedures, the estimations of BEP in Fig. 4a, Fig. 4b, Fig. 4c, and Fig. 5c can be calculated as (0.004 ± 0.001) , (0.0012 ± 0.0006) , (0.0006 ± 0.0004) , and (0.011 ± 0.005) , respectively.

We have added the above values in the Manuscript page 3 (lines 59-60), page 5 (lines 132-133), page 6 (lines 162-163, lines 167-168), and page 7 (lines 185-186, lines 205-206).

Comment 3. *“What is the maximum bit rate of the receiver? What is the limitation? If the limitation is the CMOS camera, what is the limiting bandwidth of the NV sensors themselves, and what causes that limitation? How does this bandwidth compare to currently used receivers, and how well does it fit with the vision of utilising this receiver for 6G communications?”*

Response:

We would like to clarify the issues from the following aspects:

1) Current bit rate and limitations: In our current implementation, the bit rate is limited to 50 bps due to hardware constraints. Specifically: *i) Camera limitation:* Our camera (Teledyne Photometrics Evolve 512 Delta) is limited to a maximum speed of 67 frames per second (~15

ms acquisition time per frame for full-pixel readout). Combined with the ~ 5 ms required to change microwave parameters, the total acquisition time can only be reduced to ~ 20 ms, corresponding to a maximum frame rate of 50 Hz. *ii) Microwave source limitation:* The microwave source (Windfreak Technology Synth HD) requires over 5 ms to adjust microwave parameters, adding an additional bottleneck to the system's speed. However, this aspect pertains to the transmitter and is not a factor in our receiver's design, even though it constrains the demonstration performance.

2) Potential improvements: Several improvements can dramatically increase the bit rate: *i) High-speed detectors:* Event cameras, capable of high-speed ODMR imaging, can achieve data acquisition rates up to ~ 10 kHz [R3]. Alternatively, integrating nanodiamonds on chips [R4] and collecting fluorescence signals with single-shot detectors such as photodiodes (PD), avalanche photodiodes (APD), or photomultiplier tubes (PMT) can achieve acquisition rates up to 100 kHz [R5, R6]. *ii) Optimized microwave sources:* Commercial microwave sources designed for communication applications can significantly reduce the time needed to adjust microwave parameters, enabling faster signal generation and transmission.

3) Intrinsic limitations of NV sensors: Once hardware constraints are addressed, the bit rate will be fundamentally limited by the interaction time between the microwave field and the NV center spin. For continuous-wave (CW) ODMR, the NV spin polarization time requires sufficient microwave application time to observe contrast in fluorescence. This is constrained by the transition rate between energy levels. Specifically, it refers to the k_{s0} parameter shown in Response Fig. 3, with a limited rate of 0.98 ± 0.31 MHz [R7]. This rate determines the bit rate, implying that the acquisition rate will be limited to approximately a maximum of 1 megasymbols per second in a solid-state quantum system. Achieving such speeds would require faster detectors, such as high-speed cameras or single-shot detectors like photodiodes, avalanche photodiodes, or photomultiplier tubes. In addition, achieving high data rates depends on the quality of the nanodiamonds. High-quality nanodiamonds with a large number of NV centers are critical to generating sufficiently bright fluorescence signals. With adequate brightness, the effective SNR would increase, achieving the shortest acquisition time (around 1 μ s) and the highest symbol rate.

Response Fig. 3. A diagram of energy levels with detailed transition paths.

4) Comparison to current receivers: According to the standards of 5G new radio (NR) in 3GPP [R8], a radio frame is fixed at 10 ms and consists of 10 subframes, each of which is 1 ms long. Each subframe has 1–16 slots, each of which consists of 14 symbols. Therefore, the symbol duration is 4.46–71.43 μs , corresponding to the symbol rate of 0.014–0.22 mega-symbols per second. We can observe that the symbol rate of our FND-receiver (1 mega-symbols per second) is larger than the traditional receiver in 5G NR standards (0.014–0.22 mega-symbols per second).

5) Suitability for 6G networks: Since 6G is still under development and lacks standardization, we have shown the capability of NV-based receiver in terms of symbol rate exceeds the current receivers for 5G NR standards. In addition, we would also clarify that different 6G application scenarios have distinct requirements in terms of latency, data rate, bandwidth, etc. Our system is specifically designed to target the scenario of ubiquitous radio access in the 6G networks. In comparison to state-of-the-art communication systems, our NV center-based receiver system shows significant promise for integration into 6G networks. Traditional receiver designs are limited in size due to the requisite half-wavelength spacing between antenna elements, and restricted bandwidth which necessitates multiple RF receivers for multi-band signals detection. Our design addresses these issues by offering the key advantages: NV centers operate in the quantum domain and exhibit greater sensitivity than traditional metal antennas, have no spacing constraints. This makes them promising candidates for developing compact and sensitive receivers. Moreover, in contrast to traditional receivers with components like RF antennas, analog filters, and mixers, NV-based receivers or sensors mainly use optical components for detection and down-conversion, marking them inherently immune to electrical noise. In

summary, the key advantages include: *i) Wide bandwidth*: The system can naturally support millimeter-wave bands and, with can extend into terahertz frequencies using a frequency mixer. *ii) Quantum properties*: NV centers offer unique advantages, such as high sensitivity and low noise, which could provide an edge over classical systems in specific applications.

We have added a brief discussion in Manuscript page 8 and page9 (lines 238-249), and a comprehensive discussion of modulation depth and its implications for dynamic range in the Supplementary Note 4 (pages 15-17) and Supplementary Note 7 (pages 20-21).

References:

[R3] Z. Du, M. Gupta, F. Xu, K. Zhang, J. Zhang, Y. Zhou, Y. Liu et al. "Widefield Diamond Quantum Sensing with Neuromorphic Vision Sensors." *Advanced Science* 11, no. 2 (2024): 2304355.

[R4] H. Siampour, S. Kumar, V. A. Davydov, L. F. Kulikova, V. N. Agafonov, and S. I. Bozhevolnyi. "On-chip excitation of single germanium vacancies in nanodiamonds embedded in plasmonic waveguides." *Light: Science & Applications* 7, no. 1 (2018): 61.

[R5] Q. Gu, L. Shanahan, J. W. Hart, S. Belser, N. Shofer, M. Atature, and H. S. Knowles. "Simultaneous nanorheometry and nanothermometry using intracellular diamond quantum sensors." *ACS Nano* 17, no. 20 (2023): 20034-20042.

[R6] J. Yun, K. Kim, S. Park, and D. Kim. "Temperature Selective Thermometry with Sub-Microsecond Time Resolution Using Dressed-Spin States in Diamond." *Advanced Quantum Technologies* 4, no. 11 (2021): 2100084.

[R7] J. Klatzow, J. N. Becker, P. M. Ledingham, C. Weinzetl, K. T. Kaczmarek, D. J. Saunders, J. Nunn, I. A. Walmsley, R. Uzdin, and E. Poem. "Experimental demonstration of quantum effects in the operation of microscopic heat engines." *Physical Review Letters* 122, no. 11 (2019): 110601.

[R8] 3GPP TS 38.211 – Table 4.3.2-1.

Comment 4. “*What is the sensitivity of the receiver? A plot of BER vs signal strength would be very instructive for some or all use cases.*”

Response:

For an FND-receiver, nanodiamonds are utilized to detect microwave magnetic fields. In our setup, the sensitivity is determined as follows. We conducted ten experiments to calculate the

sensitivity of FNDs when ODMR is applied for magnetic field detection. The frequency difference, denoted as Γ , between one peak and the center frequency was measured, as illustrated in Response Fig. 4. The measurement gives the result that $\Gamma = 5.15 \pm 0.127$ MHz. Given the gyromagnetic ratio (γ) being 28 GHz/T, the measurement limit of the magnetic field intensity (ΔB) can be calculated as $\Delta B = \Delta\Gamma / \gamma = (0.127 \text{ MHz}) / (28 \text{ GHz/T}) = 4.536 \mu\text{T}$. The total ODMR measurement time was 38.099 s for whole 81 frequency points scan. Consequently, the sensitivity is determined using the equation $\eta_B = \Delta B / \sqrt{t} = (4.536 \mu\text{T}) / \sqrt{(38.099 \text{ s})} = 0.735 \mu\text{T} \cdot \text{Hz}^{-1/2}$.

Response Fig. 4. Measurements of FND sensitivity. **(a)** Microscopic image of FNDs, with the spots highlighted by the red rectangle selected for sensitivity analysis. **(b)** Key parameters utilized in the sensitivity calculation process.

Furthermore, this sensitivity has a theoretical limit value [R9]:

$$\eta_B \approx 0.77 \frac{h}{g\mu_B} \frac{2\Delta\nu}{C\sqrt{R}}$$

where μ_B is the Bohr magneton, h is Planck constant, $g \approx 2$ for NV centers, C is the ODMR contrast, $\Delta\nu$ is the half-width at half maximum (HWHM), and R is photon counts. The contrast and HWHM are related to both the laser power and microwave power. We conducted experiments under varying laser and microwave power levels and plotted a map to analyze how these parameters influence the sensitivity. Because the camera intensity is not directly measured in photon counts, we cannot directly use the aforementioned formula to calculate the exact theoretical sensitivity. However, camera intensity is proportional to photon counts and thus there is a relationship for conversion between the two (though this relationship is difficult to determine precisely). We use the sum of the region of interest (ROI) intensity, I , in images

as a substitute for R . We then plotted the map between $\Delta\nu/(C\sqrt{I})$ and the laser and microwave power, as shown in Response Fig. 5.

Response Fig. 5. Sensitivity dependence on laser and microwave powers for FNDs. **(a)** Laser power dependent ODMR, measured at a microwave power of -10 dBm from the microwave source. **(b)** Microwave power dependent ODMR, performed at a laser power of 19 mW. **(c)** Relationship between ODMR contrast/ half-width at half maximum (HWHM) $\Delta\nu$ and laser power in (a). The ODMR contrast initially increases and then decreases as the optical power grows, while the linewidth decreases with respect to the increasing optical power. **(d)**

Relationship between ODMR contrast/ HWHM $\Delta\nu$ and microwave power in (b). Higher microwave power increases the ODMR contrast and broadens the linewidth. (e) Sensitivity map as a function of varying laser and microwave powers for the left peak. (f) Sensitivity map as a function of varying laser and microwave powers for the right peak.

This sensitivity can be influenced by several factors, including the microwave amplitude [R9] and excited laser power [R10]. The microwave amplitude is typically user-defined and maintained at a fixed power, making the laser power a more crucial factor in optimizing sensitivity. For laser power, the contrast initially increases with rising laser power until optical saturation pumping is reached. Beyond this point, the contrast decreases at higher laser powers [R9]. Additionally, increasing laser power narrows the linewidth and enhances the fluorescence signal. Therefore, the sensitivity improves with increasing laser power up to the saturation point. Beyond the saturation point, a balance must be achieved between contrast, linewidth, and fluorescence photoluminescence to maintain optimal sensitivity.

Furthermore, we conducted additional experiments to evaluate the relationship between BER and signal strength (microwave power). The plot of BER as a function of signal strength is shown in Response Fig. 6. From the results, it is evident that higher signal strength consistently leads to a lower BER. This is because stronger microwave signals enhance the SNR, improving the system's ability to distinguish between different symbols and reducing the likelihood of errors in symbol detection.

Response Fig. 6. The relationship between signal strength (microwave power) and BER.

We have mentioned the sensitivity value in Manuscript page 3 (lines 78-81) and added the discussion on sensitivity in the Supplementary Note 4 (pages 15-17). We have also added the

discussion on the relationship between signal strength and BER in the Manuscript page 8 (lines 228-233), while the Response Fig. 6 has been placed as Supplementary Fig. 10 (page 14).

References:

[R9] A. Dréau, M. Lesik, L. Rondin, P. Spinicelli, O. Arcizet, J-F. Roch, and V. Jacques. "Avoiding power broadening in optically detected magnetic resonance of single NV defects for enhanced dc magnetic field sensitivity." *Physical Review B—Condensed Matter and Materials Physics* 84, no. 19 (2011): 195204.

[R10] K. Jensen, V. M. Acosta, A. Jarmola, and D. Budker. "Light narrowing of magnetic resonances in ensembles of nitrogen-vacancy centers in diamond." *Physical Review B—Condensed Matter and Materials Physics* 87, no. 1 (2013): 014115.

Comment 5. *“Is the bit detection process implemented in real-time or on pre-collected data? It seems very resource inefficient to need to process a 12.2 megapixel image for each bit. What is the maximum bitrate this sort of process is capable of maintaining in real time with suitable/specialised hardware?”*

Response:

To begin with, we clarify that the bit detection process in our demonstration is performed on pre-collected data. However, we would like to clarify that, instead of processing an entire 12.2-megapixel image for each bit, we only process an intensity image of size 512×512 pixels. This significantly reduces the computational load, and the process is not time-consuming. With the use of suitable or specialized hardware, such as GPUs or FPGAs, this method could potentially achieve real-time processing. Specifically, the bit detection refers to comparing the received fluorescence intensity image to the reference intensity images by computing the mean square error (MSE). We discuss the following two kinds of hardware.

1) GPU: Consider the utilization of GPU like the NVIDIA A100, which can deliver up to 19.5 TFLOPs for single-precision (FP32) calculations, resulting in the computing speed of 19.5×10^{12} FLOPs/second. To compute the MSE between two matrices of size 512×512 , the computation overhead, measured in floating-point operations (FLOPs) is estimated as follows:

- Subtraction: Compute the difference between corresponding elements of the two matrices. Each of the 512×512 elements requires one subtraction operation. (Total: $512 \times 512 = 262,144$ FLOPs)

- Squaring: Square each element of the resulting difference matrix. Each of the 512×512 elements in the difference matrix needs to be squared. (Total: $512 \times 512 = 262,144$ FLOPs)
- Summation: Sum all the squared elements. Summing 262,144 elements can be performed in a binary tree-like fashion with 262,143 additions (since summing n elements requires $(n - 1)$ additions). (Total: 262,143 FLOPs)
- Division: Divide the sum by the total number of elements. This operation is optional and requires only 1 FLOP.

Therefore, a total of 786,432 FLOPs are needed to compute the MSE for bit-pair detection. Hence, the processing time required to perform 786,432 FLOPs for bit detection is 40 nanoseconds. In our settings, there are four reference fluorescence patterns for comparison and each detection process recovers two bits. Thus, the processing latency can be estimated as $40 \times 4 / 2 = 80$ nanoseconds/bit. In this context, the bit rate is up to 12.5 Mbps.

2) FPGA: Consider the utilization of the Intel Stratix 10 FPGA, which can operate at clock frequency up to 1 GHz (1 nanosecond per clock cycle). We can leverage the FPGA's capability to process the difference of two 512×512 matrices in parallel. This guarantees the maximum parallelism and minimize processing time. The computation time is calculated as follows:

- Element-wise subtraction and squaring: Each subtraction and squaring operation can be done in parallel in one clock cycle. (Total: 1 clock cycle)
- Parallel accumulation: The accumulation is done in a tree-like structure: In the first level, 262,144 elements are reduced to 131,072 partial sums; In the second level, 131,072 partial sums are reduced to 65,536 sums; Continue this process until we get the final sum. Hence, the number of levels in the reduction tree is $\log_2 262144 = 18$ levels. (Total: 18 clock cycles)
- Division: The optional division operation takes one clock cycle.

Therefore, a total of 20 clock cycles are needed to compute the MSE for bit-pair detection. Hence, the processing time is 20 nanoseconds. In our settings, the bit rate is up to 25 Mbps. To further enhance the bit rate, more complex modulation schemes with higher constellation order should be adopted such that each symbol contains more bits. From this perspective, the processing speed can be characterized as 25 mega-symbols per second, which is faster than the symbol rate limit (1 mega-symbols per second) as discussed in the response to Comment 3. Hence, the real-time processing can be guaranteed using the hardware.

Comment 6. *“It was not very clear to me why the number of channels was limited to a*

maximum of 5, or why the utilisation rate is limited to 5/45. What is the estimation that has occurred to reach 5 users? Perhaps I have missed this discussion or estimation in the paper, but I think it is worthy of at least brief discussion in the main text and detailed discussion in the supplementary.”

Response:

We would like to clarify this issue from the three perspectives:

1) Key factors affecting the number of channels: The number of channels (or users) in this system is determined by the three factors. *i) Magnetic field gradients:* The applied magnetic field gradient is stated to be approximately $0.023 \text{ G}/\mu\text{m}$. This gradient causes a shift in the ODMR frequency of about $1 \text{ MHz}/\mu\text{m}$. *ii) Field of view (FOV):* The system’s FOV is $75 \mu\text{m} \times 75 \mu\text{m}$, which defines a region where FNDs can be observed simultaneously. *iii) Frequency separation of Lorentzian peaks:* To avoid overlap in the ODMR spectrum, the Lorentzian peaks of different FNDs must be distinct. The separation between peaks depends on the axial orientation of the FNDs relative to the magnetic field gradient.

2) How 5 users were estimated: The estimation of 5 users arises from the ability to resolve distinct ODMR peaks within the given FOV and magnetic field gradient: In the experiment, we can observe 45 FNDs within the $75 \mu\text{m} \times 75 \mu\text{m}$ FOV. The magnetic field gradient of $0.023 \text{ G}/\mu\text{m}$ results in an approximate ODMR shift of $1 \text{ MHz}/\mu\text{m}$. For an FND to be resolvable in frequency space, its Lorentzian peak must not overlap with other peaks. This requires a minimum frequency separation of at least 1 MHz between neighboring FNDs. The distinct axial orientations of FNDs contribute to differences in their resonance frequencies. However, due to the randomness in axial orientations, not all FNDs in the FOV will produce sufficiently separated peaks. Out of the 45 visible spots in the FOV, only 5 FNDs are sufficiently distinct in frequency (due to spatial separation and axial orientation differences) to avoid peak overlap. This results in a utilization ratio of $5/45$, meaning only 5 out of 45 available positions are usable for communication in this setup.

3) Why the utilization rate is limited to 5/45: The utilization rate is limited due to the physical and technical constraints of the system: *i) Field of view constraints:* The FOV restricts the number of observable FNDs to 45. Within this small area, the magnetic field gradient causes shifts in the ODMR peaks, but not all FNDs will have sufficiently distinct frequencies. *ii) Frequency resolution requirements:* The requirement to avoid overlapping Lorentzian peaks reduces the number of distinguishable FNDs. Even though there are 45 potential spots, only 5

FNDs meet the criteria for distinct peak separation. *iii) Axial orientation limitations:* The distinct resonance frequencies of FNDs arise from their unique axial orientations. However, the variation in these orientations is not unlimited, restricting the number of usable FNDs.

We have mentioned this issue in the Manuscript page 7 (line 188, lines 190-192), and added a comprehensive discussion to clarify it in the Supplementary Note 5 (page 18).

Comment 7. “*What is the size, weight, and power consumption of this whole receiver design? What can be done to improve these and what is the potential and limitations in SWaP-C for large scale deployment?*”

Response:

Here are the details of the size, weight, and power consumption (SWaP-C) of our implemented receiver.

1) SWaP-C breakdown and improvements:

i) Size: Our current receiver prototype measures 300 mm × 300 mm × 200 mm, encompassing all optical and electronic components. The size can be significantly reduced by integrating the optical and electronic components into a chip-scale platform [R11], minimizing reliance on bulky breadboards and external devices. On-chip integration of FNDs and fiber-coupled excitation of NV centers would make the system more portable and scalable.

ii) Weight: The total weight is approximately 8 kg, with optical breadboards contributing 6 kg and the remaining optical and electronic components weighing below 2 kg. Replacing heavy breadboards with compact, purpose-built optical mounts or chip-scale photonic devices would drastically reduce weight. With full on-chip integration and the removal of breadboards, the weight can be greatly reduced, enabling deployment in portable or even handheld devices.

iii) Power consumption: In our receiver prototype, the power consumption for the camera and laser are 1.17 W and 0.7 W respectively, resulting in a total power consumption of 1.87 W. Replacing the camera with more efficient optical detectors can reduce power consumption significantly. Advanced laser designs with lower power thresholds can further reduce optical component power requirements. Integrating FNDs on-chip can operate efficiently with laser power levels around 100 μW [R11].

2) Potential: *i) Miniaturization:* On-chip integration of FNDs and fibers would allow for highly compact, lightweight, and low-power modules suitable for large-scale deployment in portable devices and networks. *ii) Cost reduction:* Once integrated into scalable semiconductor

or photonic platforms, the cost of manufacturing and deploying these devices would decrease significantly. *iii) Versatility:* The system's ability to operate at low power and detect weak signals makes it ideal for applications in ubiquitous radio access wireless systems.

3) Limitations: *i) Fabrication challenges:* Incorporating FNDs into photonic integrated circuits while maintaining high operational efficiency and sensitivity requires precise fabrication techniques. *ii) Material availability and processing:* Integration of FNDs into scalable platforms (e.g., silicon photonics) might require additional material-processing steps, increasing complexity. *iii) Advanced laser design:* The reduction of power consumption requires advanced laser designs with lower power thresholds, which is challenging to maintain the quality of laser at the same time.

We have added the discussion in the Manuscript page 10 (lines 280-286) and Supplementary Note 9 (page 25).

Reference:

[R11] H. Siampour, S. Kumar, V. A. Davydov, L. F. Kulikova, V. N. Agafonov, and S. I. Bozhevolnyi. "On-chip excitation of single germanium vacancies in nanodiamonds embedded in plasmonic waveguides." *Light: Science & Applications* 7, no. 1 (2018): 61.

Comment 8. "*Some minor details:*

Line 41, "traditional receivers consists of a series...", should read "traditional receivers that consist of a series..."

Line 42, marking should be making."

Response:

We sincerely thank you for your careful reading. We have corrected the typos as you pointed out. The whole manuscript has been carefully proofread as best as we can.

Reviewer #4

“I co-reviewed this manuscript with one of the reviewers who provided the listed reports. This is part of the Communications Engineering initiative to facilitate training in peer review and to provide appropriate recognition for Early Career Researchers who co-review manuscripts.”

Response:

We sincerely appreciate the reviewer’s effort on our paper. All the concerns from the above reviewers are addressed in detail in the point-to-point response.

Reviewers' comments:

Reviewer #1 (Remarks to the Author):

The authors have adequately addressed all my comments.

Reviewer #3 (Remarks to the Author):

The authors have improved the manuscript that neatly combines quantum sensors for novel RF applications. I can recommend this article to be published without further adjustments.

Reviewer #4 (Remarks to the Author):

The authors have significantly improved the manuscript and I think it is a novel and interesting paper for the fields of quantum sensing, and RF communications. I can recommend this work for publication without any further peer review. I only have a couple of minor suggestions to improve the paper further:

Line 79 – “based on the such contrast” should read “based on this contrast”
Supplementary reference 10 is showing up as incomplete.

Supplementary Note 8, 6.) Minimum detected power, minimum detected power is quoted from the microwave source. This should be instead estimated and stated at the receiver based on the microwave amplification and transmission chain. A number that is equivalent to the reference signal received power (RSRP) number quoted in 5G comms should be presented as the minimum detected power. To me this is the number 1 concern for these type of receivers. Typical 5G receivers operate with received power in the -60 to -90dBm range. How many orders of magnitude away from this range is this receiver currently, and it should be stated how much each of the proposed improvements can achieve. A multi-access receiver still has utility even if its minimum received power for operation is higher than other single access alternatives but if there will be a trade-off it should be quantified.

Reviewer #1

“The authors have adequately addressed all my comments.”

Response:

We are grateful for the reviewer’s the positive comment.

Reviewer #3

“The authors have improved the manuscript that neatly combines quantum sensors for novel RF applications. I can recommend this article to be published without further adjustments.”

Response:

We thank the reviewer for the positive comment and recommendation.

Reviewer #4

“The authors have significantly improved the manuscript and I think it is a novel and interesting paper for the fields of quantum sensing, and RF communications. I can recommend this work for publication without any further peer review. I only have a couple of minor suggestions to improve the paper further:”

Response:

We appreciate the reviewer’s positive comments and recommendation.

Comment 1. *“Line 79 – “based on the such contrast” should read “based on this contrast” Supplementary reference 10 is showing up as incomplete.”*

Response:

We sincerely thank the reviewer for careful reading. We have corrected the typo and provided the complete reference as pointed out.

Comment 2. *“Supplementary Note 8, 6.) Minimum detected power; minimum detected power is quoted from the microwave source. This should be instead estimated and stated at the receiver based on the microwave amplification and transmission chain. A number that is equivalent to the reference signal received power (RSRP) number quoted in 5G comms should be presented as the minimum detected power. To me this is the number I concern for these type*

of receivers. Typical 5G receivers operate with received power in the -60 to -90dBm range. How many orders of magnitude away from this range is this receiver currently, and it should be stated how much each of the proposed improvements can achieve. A multi-access receiver still has utility even if its minimum received power for operation is higher than other single access alternatives but if there will be a trade-off it should be quantified.”

Response:

We fully understand the importance of estimating and presenting the receiver’s sensitivity, as this is a critical parameter for evaluating the performance of our system. Below, we address each aspect of the reviewer’s comment, incorporating the experimental results from our study.

1) Estimating the Minimum Detected Power at the Receiver: In our experiments, the received power at the nanodiamonds was carefully measured and clarified. Below, we summarize the key findings:

- When the microwave power is set to -25 dBm, it is amplified to 5 dBm at the input of the microwave structure. The gold line within the microwave structure will emit -31 dBm of power, with the remaining power being transmitted to the end of the structure and some energy being lost as heat. This results in an emitted power of -31 dBm. After accounting for transmission losses, the nanodiamonds receive a power of -47.2 dBm.
- When the microwave power is set to 0 dBm, it is amplified to 30 dBm at the input of the microwave structure. The gold line within the microwave structure will emit -6 dBm of power, with the remaining power being transmitted to the end of the structure and some energy being lost as heat. This results in an emitted power of -6 dBm. After transmission, the nanodiamonds receive a power of -22.2 dBm.

Thus, the received power in our experiments ranges from -47.2 dBm to -22.2 dBm, depending on the input microwave power and the system configuration.

2) Comparison with 5G RSRP Range: The typical received power in 5G communications, as the reviewer noted, ranges from -60 dBm to -90 dBm. In comparison, the received power in our experiments ranges from -47.2 dBm to -22.2 dBm, which is higher than the 5G RSRP range. This indicates that the receiver currently requires a higher minimum received power for operation compared to typical 5G receivers. The difference between our receiver’s detection and the 5G RSRP range is approximately 13 dB to 67 dB, depending on the specific conditions.

3) Proposed Improvements: To address the gap, adding a low-noise amplifier (LNA) at the receiver’s front end can significantly enhance sensitivity by reducing the system’s effective

noise figure (NF). The noise floor can be lowered by using a high-gain, low-NF LNA, such as a 30 dB gain, 1 dB NF LNA, which can improve the SNR by 5–10 dB, depending on initial system losses. Advanced signal processing techniques like matched filtering can further improve SNR by 10–20 dB by suppressing noise and enhancing signals. Combined, the noise floor can be reduced by 15–30 dB, bringing performance closer to the 5G RSRP range.

4) Utility of the Multi-Access Receiver: We appreciate the reviewer’s comment regarding the utility of the multi-access receiver, even if its sensitivity is higher than that of single-access alternatives. As highlighted in the manuscript, the multi-access capability of our system provides unique advantages, such as the ability to simultaneously process signals across multiple channels or frequencies. These features are particularly advantageous for applications such as multi-band communication and ubiquitous radio access.

The discussions have been added to the caption of Supplementary Fig. 10 (Supplementary Information, Page 14).